# Cell-type specific profiling of histone post-translational modifications in the adult mouse striatum

Marco D. Carpenter[1,2,3,5], Delaney K. Fischer[1,2,3,5], Shuo Zhang[1,2,3,5], Allison M. Bond[4], Kyle S. Czarnecki[1], Morgan T. Woolf[1], Hongjun Song [3,4] & Elizabeth A. Heller [1,2,3] ✉

Epigenetic gene regulation in the heterogeneous brain remains challenging to decipher with current strategies. Bulk tissue analysis from pooled subjects reflects the average of cell-type specific changes across cell-types and individuals, which obscures causal relationships between epigenetic modifications, regulation of gene expression, and complex pathology. To address these limitations, we optimized a hybrid protocol, ICuRuS, for the isolation of nuclei tagged in specific cell-types and histone post translational modification profiling from the striatum of a single mouse. We combined affinity-based isolation of the medium spiny neuron subtypes, Adenosine 2a Receptor or Dopamine Receptor D1, with cleavage of histone-DNA complexes using an antibody-targeted micrococcal nuclease to release DNA complexes for paired end sequencing. Unlike fluorescence activated cell sorting paired with chromatin immunoprecipitation, ICuRuS allowed for robust epigenetic profiling at cell-type specific resolution. Our analysis provides a framework to understand combinatorial relationships between neuronal-subtype-specific epigenetic modifications and gene expression.

The epigenome determines cellular identity and serves as the interface between genes and environment to determine individual susceptibility to disease. The brain transcriptome and epigenome are cell-type specific, underscoring the diversity and specialization of intermingled neuronal and glial subtypes[1]. A specific cell-type may contain intrinsic disease-causing factors or be one of many individual cell-types underlying a complex pathology[2]. For example, in Parkinson's disease, degeneration specifically of dopaminergic neurons perturbs additional cell-types, including medium spiny neurons (MSNs)[3]. Genes implicated in schizophrenia are expressed specifically in MSNs, rather than interneurons, astrocytes, or glia[4]. With respect to reward pathophysiology, a robust literature implicates A2a- and D1-MSN specific connectivity[5], gene expression[6] and chromatin accessibility[7]. In

addition to cell-type specificity, transcriptomic profiling underscores individual subject variation in pathogenic gene expression associated with several brain disorders[8]. While methods are emerging to profile cell-type specific features of the epigenome[1,9,10], few examine individual subject variability[9] and none report MSN-specific hPTM profiling from a single animal.

Chromatin immunoprecipitation sequencing (ChIP-seq) has been widely used to profile hPTMs in specific neuronal populations but this method requires large numbers of nuclei and is prone to high background[11,12]. We developed ICuRuS by combining methods for isolation of nuclei tagged in specific cell-types[10] (INTACT) and hPTM profiling by cleavage under targets & release using nuclease[12,13] (CnR), followed by next generation sequencing. These methods were highly

[1]Department of Systems Pharmacology and Translational Therapeutics, University of Pennsylvania, Philadelphia, PA, USA. [2]Institute for Translational Medicine and Therapeutics, University of Pennsylvania, Philadelphia, PA, USA. [3]Penn Epigenetics Institute, Perelman School of Medicine, University of Pennsylvania, Philadelphia, PA, USA. [4]Department of Neuroscience, University of Pennsylvania, Philadelphia, PA, USA. [5]These authors contributed equally: Marco D. Carpenter, Delaney K. Fischer, Shuo Zhang. ✉e-mail: eheller@pennmedicine.upenn.edu

amenable to a combined approach, since INTACT culminates in bead-immobilized nuclei, which are the initial substrate in CnR[12,13]. This approach addressed the following limitations of ChIP-seq for low-input epigenetic profiling of a single cell-type from a single brain region of a single subject: (1) The reliability and detection sensitivity of subtle, physiologically-relevant changes is limited[11]. This necessitates pooling across subjects, which hinders downstream correlations between hPTMs and individual subject behavior and/or pathology[1]. (2) Cellular integrity is impacted by the nuclear isolation technique[14]. For example, fluorescence-activated cell sorting (FACS) leads to ectopic upregulation of activity-dependent genes[14–16], cellular stress, artifactual DNA shearing, and increased background[17]. To compensate for high background, data is normalized to input chromatin, requiring greater sequencing costs. In addition, FACS is limited by efficiencies and availabilities of cell-type specific nuclear antibodies. (3) Formaldehyde fixation masks protein epitopes and stabilizes highly transient chromatin binding, leading to false positive IP[18]. Native ChIP does not require cross-linking and is an efficient method of hPTM profiling but transcription factor profiling is limited by insufficient binding efficiencies[13] and the required sequencing depth for native ChIP is 10x that required for CnR. ICuRuS overcomes these obstacles and allows for low-cost sequencing with limited starting material.

Regulation of hPTMs enrichment permits transcription factor binding necessary for cell-type specific gene expression. Permissive and repressive hPTMs, H3K4me3 (histone H3 lysine 4 tri-methylation) and H3K27me3, define a "poised" chromatin state that has traditionally been referred to as developmentally poised as opposed to poised for neural activity-dependent stimulation[19]. In this context, loss of H3K4me3/H3K27me3 co-enrichment to H3K4me3 enrichment alone is positively associated with gene induction during neuronal differentiation[20]. However, similar mechanisms may be at play in postmitotic neurons, for stable, long-term changes in gene expression[21]. Here, we characterized cell-type specific hPTM profiles in striatal MSNs. First, we isolated A2a and D1 nuclei from striatum of sufficient number, specificity, and quality for epigenomic profiling. To model and predict cell-type-specific gene expression patterns, we profiled both permissive and repressive hPTMs, H3K4me3 and H3K27me3, respectively, which mark expressed genes in MSNs. We showed A2a and D1 nuclei differ in their relative enrichment of H3K4me3 and H3K27me3 at cell-type specific genes but are largely similar in genome-wide enrichment. MSN-subtype specific gene expression was defined by MSN-subtype specific enrichment of H3K4me3 or H3K27me3 or both. We highlighted our findings on *Egr3*, a gene relevant to substance use disorder and activated specifically in D1 nuclei in response to cocaine exposure[22]. We report here that the *Egr3* promoter was enriched in H3K4me3 in both A2a and D1 MSNs but depleted in H3K27me3 specifically in D1 MSNs relative to the opposing cell-type. Overall, high-resolution epigenomic profiles generated by ICuRuS defined combinatorial relationships between neuronal-subtype-specific hPTMs and gene expression.

## Results

### INTACT purified A2a and D1 MSN nuclei from a single mouse brain

To generate a mouse line for INTACT affinity purification of striatal A2a and D1 nuclei, we crossed the established SUN1-sfGFP-Myc mouse line[10], expressing a GFP-affinity-tagged SUN1 nuclear receptor under the control of a loxP-3xPolyA-loxP transcriptional stop cassette, to either A2a- or D1-Cre mouse lines. A2a-Cre; or D1-Cre; SUN1-GFP mice were healthy, fertile and displayed no phenotypic abnormalities (Supplementary Fig. 1A). Double immunohistochemistry of GFP and A2a or GFP and D1 showed expression of nuclear SUN1-GFP in the target cell-type (Fig. 1A, B). Using INTACT[10], we isolated A2a or D1 nuclei from the striatum of a single mouse with an anti-GFP antibody (Fig. 1C, D). Fluorescence microscopy showed a sufficient number of

nuclei (8000 -10,000) were recovered for downstream hPTM profiling by CnR (Fig. 1E)[13,23–25] and mRNA quantification by qPCR of high quality mRNA (Supplementary Fig. 1B). To validate the specificity of the isolated nuclei, we measured mRNA of cell-type specific genes, *A2a* and *Drd1*, in the affinity purified and 'flow-through' fractions after INTACT of each cell-type (Fig. 1F, G). Following A2a INTACT, *A2a* mRNA was enriched and *Drd1* mRNA was depleted in the A2a affinity purified fraction relative to the flow-through (Fig. 1F). Following D1 INTACT, *Drd1* mRNA was enriched and *A2a* mRNA was depleted in the D1 affinity purified fraction relative to the flow-through (Fig. 1G). These data showed that INTACT successfully isolated A2a and D1 nuclei from a single mouse striatum.

### Epigenomic profiling using CnR in N2a Cells

We first validated H3K4me3 and H3K27me3 CnR in N2a cells[26,27] (Supplementary Fig. 2A-J) finding that H3K4me3 and H3K27me3 profiles aligned with published N2a ChIP-seq data[26,27] (Supplementary Fig. 2B). For H3K4me3 and H3K27me3, the genomic read distribution across replicates was highly similar, based on Pearson's correlation coefficient (PCC; H3K4me3 PCC: 0.99, H3K27me3 PCC: 0.94; Supplementary Fig. 2C, D), and correlated with corresponding ChIP-seq data for each modification (H3K4me3 PCC: 0.78, H3K27me3 PCC: 0.63; Supplementary Fig. 2C, D), but not with IgG control (PCC < 0.1; Supplementary Fig. 2C, D). Moreover, H3K4me3 and H3K27me3 CnR reads were enriched around peaks called from corresponding N2a ChIP-seq data[26,27] (Supplementary Fig. 2E, F), indicating that CnR recapitulated features of ChIP-seq. Quantification of the fraction of reads in peaks (FRiP) revealed that H3K4me3 CnR in N2a cells produced high quality data with negligible signal to noise, similar to published ChIP-seq FRiP in this cell line (Supplementary Fig. 2G). H3K27me3 CnR FRiP also showed high signal to noise which was slightly enhanced compared to published ChIP-seq FRiP[26,27] (Supplementary Fig. 2I). There was a 63% and 35% overlap in H3K4me3 and H3K27me3 peaks, respectively, between N2a CnR and previously published ChIP-seq data[26,27] (Supplementary Fig. 2H, J). Thus, CnR successfully profiled H3K4me3 and H3K27me3 modifications in N2a cells.

### Validation of epigenomic profiling of MSNs using published ChIP-Seq

We next optimized ICuRuS by combining INTACT and CnR, to profile hPTMs in specific MSN subtypes of the mouse striatum. CnR is easily applied to neuronal nuclei following INTACT, as sfGFP-SUN1 + nuclei were immobilized during INTACT on paramagnetic beads[13]. Bead-bound nuclei were then incubated with antibodies against H3K4me3 and H3K27me3 and subjected to antibody-guided nucleosomal MNase cleavage (see Supplementary Fig. 2A), followed by next-generation sequencing (NGS) and validation by comparison to published ChIP-seq datasets. We found that A2a and D1 ICuRuS H3K4me3 and H3K27me3 profiles were similar to corresponding NAc H3K4me3 and H3K27me3 ChIP-seq profiles[28] (Fig. 2A). To further validate striatal ICuRuS, we computed correlation matrices comparing A2a and D1 ICuRuS replicates to NAc ChIP-seq data using read coverages for the entire genome in 1-kb bins[28]. Within A2a and D1 cell-types, H3K4me3 ICuRuS replicates were highly similar (PCC: 0.99 for both A2a and D1; Fig. 2B). Between A2a and D1 cell-types, H3K4me3 replicates were more similar to each other (PCC: 0.97) than to bulk striatal ChIP-Seq (PCC: 0.53-0.73; Fig. 2B). Within A2a and D1 cell-types, H3K27me3 ICuRuS replicates were highly similar (A2a PCC: 0.89; D1 PCC: .76; Fig. 2C). Between A2a and D1 cell-types, H3K27me3 ICuRuS replicates were more similar to each other (PCC: 0.76 – 0.77) than to bulk striatal ChIP-Seq (PCC: 0.37-0.48; Fig. 2C). Next, we quantified H3K4me3 and H3K27me3 ICuRuS signal from each cell-type in peaks called from bulk NAc ChIP-seq[28]. We found that both A2a and D1 ICuRuS H3K4me3 and H3K27me3 reads were enriched at NAc ChIP-seq peak centers (Fig. 2D, E). In contrast, control ICuRuS using IgG or no antibody resulted in sparse enrichment

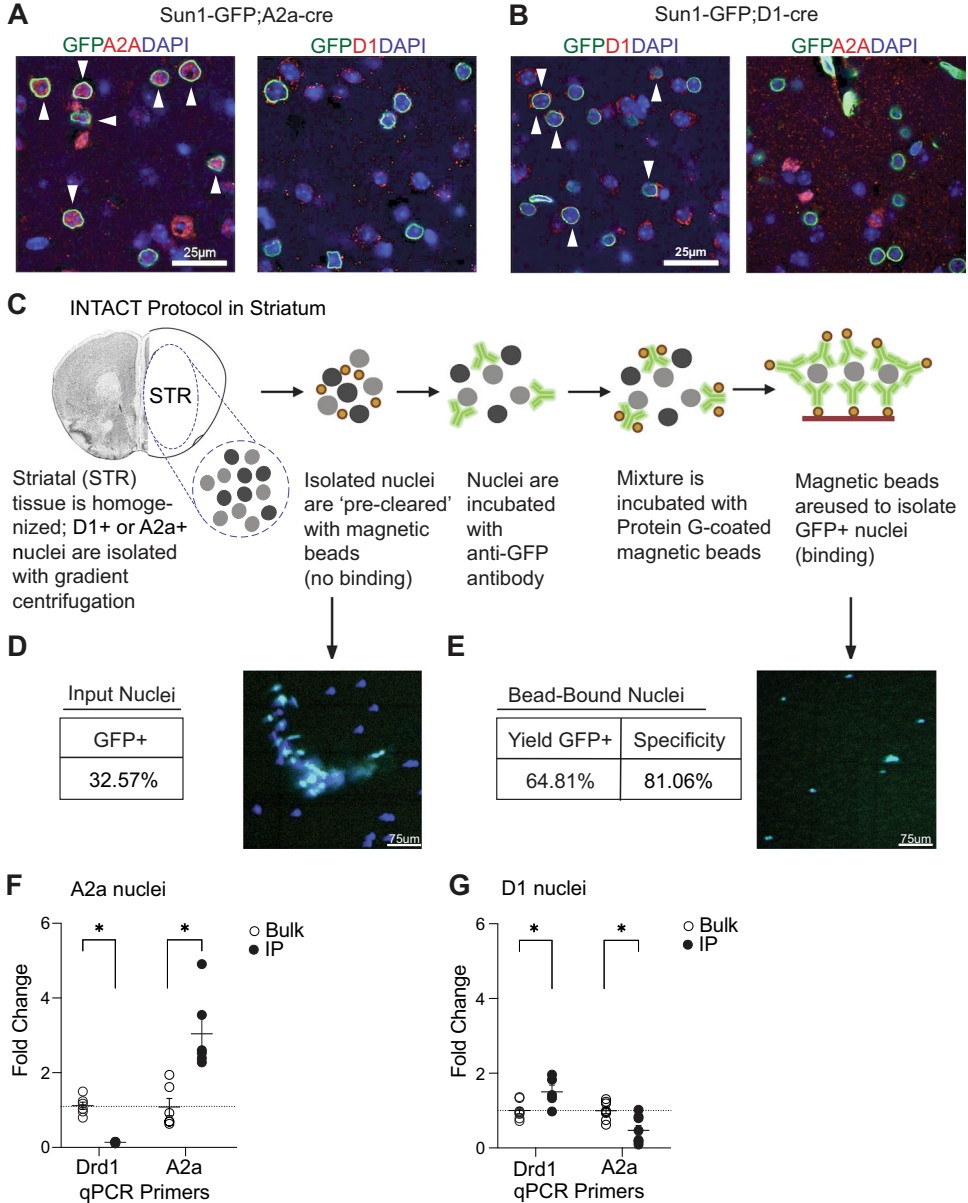

**Fig. 1 | SUN1-GFP facilitates isolation of D1 or A2a neurons in the striatum.**
**A** Immunohistochemistry displaying DAPI/GFP/A2a (left) and D1 (right) from the striatum of SUN1-GFP;A2a-CRE + animals. $n = 3$/group. Arrows indicate co-localization of all three markers. **B** Immunohistochemistry displaying DAPI/GFP/Drd1 (left) and A2a (right) from the striatum of SUN1-GFP; D1-CRE + animals. $n = 3$/group. Arrows indicate co-localization of all three markers. These experiments were replicated 3 times with similar results. **C** INTACT schematic created at Bior-ender.com. **D** Total (%) of Cre + cells in striatum. Image depicts total nuclei isolation ($n = 2$/cell type and antibody) **E** Yield (%) and specificity (%) of GFP + nuclei. Image

depicts GFP + nuclei isolation. $n = 2$/cell type and antibody. **F** mRNA validation for INTACT Sun1-GFP; A2a-Cre + vs bulk (Cre- nuclei); $n = 5$, unpaired two-tailed t test, A2a Nuclei, Drd1: (t (10) = 9.871, $p < 0.001$); A2a: (t (10) = 4.129, $P = 0.006$.), nor-malized to GAPDH. Data are presented as mean values +/- SEM *$P < 0.05$. **G** mRNA validation for INTACT Sun1-GFP; D1-Cre + vs bulk (Cre- nuclei); $n = 5$, unpaired two-tailed t test, D1 Nuclei, D1: (t (10) = 2.920, $p = 0.0278$); A2a: (t (10) = 3.421, $p = 0.012$), normalized to GAPDH. Data are presented as mean values +/- SEM. *$P < 0.05$. Source data are provided as a Source Data file.

around the peaks (Fig. 2D, E; Supplementary Fig. 2M-T). We next called ICuRuS peaks and found 78% and 42% of H3K4me3 and H3K27me3 peaks overlapped, respectively, between ICuRuS and previously pub-lished ChIP-seq data[28] (Supplementary Fig. 3A, B; Supplementary File 1) and A2a and D1 ICuRuS H3K4me3 and H3K27me3 reads populated corresponding NAc ChIP-Seq peaks at comparable levels (Supple-mentary Fig. 3C, D).

The accuracy and robustness of ICuRuS required considerable optimization. We found antibody selection to be the most important factor for successful ICuRuS (See Methods for antibody list). The same antibodies that produced robust data in ~10 K N2a cells did not pro-duce comparable data in A2a and D1 isolated nuclei (Supplementary

Fig. 2A). Specifically, H3K4me3 Antibody 3 (Ab3) or H3K27me3 Ab2 in N2a cells resulted in specific read enrichment around peaks called from N2a H3K4me3 ChIP-seq data[27] (Supplementary Fig. 2B, E, F) but nonspecific enrichment in A2a or D1 isolated nuclei (Supplementary Fig. 2M-R). The ability of the antibodies to identify the molecular target is affected not just by the presence of the target but also by neigh-boring hPTMs[29]. We hypothesize the difference in input values between H3K4me3 and H3K27me3 antibodies is related to the nature of the hPTM itself. These data highlight differences between in vitro and in vivo CnR that can be addressed by rigorous antibody selection when analyzing additional hPTMs beyond H3K4me3 and H3K27me3 in mouse brain.

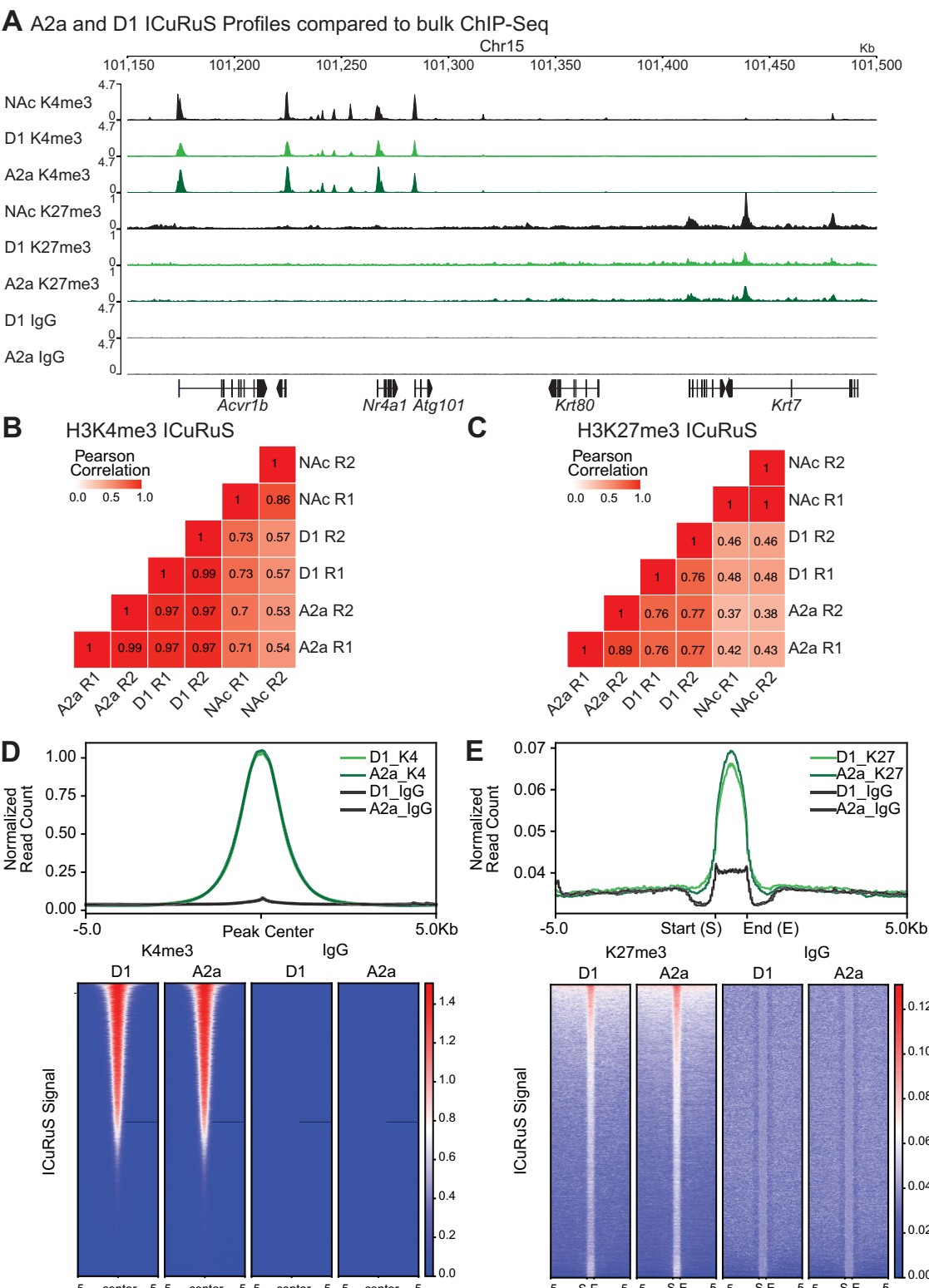

**Fig. 2 | Validation of epigenomic profiling of MSNs using published ChIP-Seq data. A** Representative genome browser views of different profiling methods and cell-types. Nucleus accumbens (NAc) bulk ChIP-seq data come from[28]. Signal is normalized to total mapped reads: Count Per Million (CPM). **B** H3K4me3 and **C)** H3K27me3 heatmap showing Pearson's correlation coefficients among NAc ChIP-seq, D1 CnR, and A2a CnR. Two replicates are shown for each dataset. The correlation coefficients were calculated by dividing the genome into 1 kb bins and counting reads in each bin. **D** Enrichment of A2a and D1 CnR H3K4me3 or IgG signal centered on peaks called from NAc H3K4me3 ChIP-seq by MACS2. Top 20,000 peaks sorted by MACS2 score were used. Heatmap shows signal around individual peaks, and the averaged signal is shown above the heatmap. Two replicates of CnR data were merged. Signal is normalized by signal and variance across transcriptionally constant genes (quantile normalization). **E** Enrichment of A2a and D1 CnR H3K27me3 or IgG signal around peaks called from NAc H3K27me3 ChIP-seq by SICER. Peaks within 5 kb were merged to avoid double counting. Heatmap shows signal around individual peaks, and the averaged signal is shown above the heatmap. Two replicates of CnR data were merged. Signal is normalized by variance across transcriptionally constant genes (quantile normalization).

To demonstrate the advantage of ICuRuS, we compared H3K4me3 and H3K27me3 ICuRuS signal to native ChIP-Seq signal from bulk excitatory neurons isolated via INTACT[10]. We found improved H3K4me3 signal to noise quantified by FRiP (Supplementary Fig. 3C, E). H3K27me3 signal was more similar between the methods (Supplementary Fig. 3D, F). However, background was higher in H3K27me3 native ChIP-seq which increases the probability of false positives (Supplementary Fig. 3F). Thus, ICuRuS provided improved and reliable high quality epigenomic profiling data across different hPTMs.

## ICuRuS validated A2a and D1 cell-type specific expression patterns

To interrogate the relationships between cell-type specific hPTMs and mRNA expression, we integrated A2a and D1 H3K4me3 and H3K27me3 profiles with published mRNA profiles of D2- and D1-specific translatosomes (i.e. translating RNA profiling by Ribo-Tag)[6]. As expected, A2a and D1 showed cell-type specific enrichment of H3K4me3 or H3K27me3, in a window 2 kb upstream and 1 kb downstream of the TSS[30], at *A2a* and *Drd1* and other cell-type specific marker genes[6]. Specifically, A2a nuclei were enriched with H3K4me3 and depleted of H3K27me3 at *A2a*, *Drd2*, and *Penk* relative to H3K4me3 and H3K27me3 enrichment in D1 nuclei, which correlates with cell-type specific expression of *A2a*, *Drd2*, and *Penk* mRNA (Fig. 3A-C). D1 nuclei were enriched with H3K4me3 and depleted of H3K27me3 at *Drd1*, *Pdyn*, and *Tac1* relative to H3K4me3 and H3K27me3 enrichment in A2a expressing nuclei, which correlates with cell-type specific expression of *Drd1*, *Pdyn*, and *Tac1* mRNA (Fig. 3D-F). Next, we called peaks at A2a and D1-specefic genes. Surprisingly, H3K4me3 peaks were called at both A2a- and D1-specific genes (Fig. 3G, H), which suggested the presence of H3K4me3 or H3K27me3 alone did not predict cell type specific gene expression. To determine if H3K4me3 or H3K27me3 ICuRuS signal correlated with cell type specific gene expression in A2a and D1 nuclei, we quantified the association between cell-type specific gene expression and levels of H3K4me3 or H3K27me3 enrichment. To accomplish this, we segmented H3K4me3 and H3K27me3 signal across three gene expression groups in A2a and D1 nuclei: high, (Fragments Per Kilobase Million, FPKM >= 10), medium (1 <= FPKM < 10) and low gene expression (FPKM < 1)[6]. A2a H3K4me3 reads were enriched in the promoter of highly expressed A2a genes (Fig. 3I) and A2a H3K27me3 reads were enriched in lowly expressed genes (Fig. 3J). Similarly, D1 H3K4me3 reads were enriched in the promoter of highly expressed D1 genes (Fig. 3K) and D1 H3K27me3 reads were enriched in lowly expressed genes (Fig. 3L). These results were consistent with the known association of H3K4me3 with gene expression and H3K27me3 with gene repression[31] and showed that H3K4me3 and H3K27me3 ICuRuS profiles corresponded with predicted hPTM enrichment based on cell-type specific gene expression. Using data normalized to counts per million, we observed that both global H3K4me3 (Supplementary Fig. 3A, C, D; $t_{(19999)}$ = −131.01, p < 2.2e-16) and global H3K27me3 (Supplementary Figure 3B, E, F; $t_{(40405)}$ = −86.189, p < 2.2e-16) signal in peaks were greater in A2a than D1, although the majority of H3K4me3 and H3K27me3 peaks were observed in both A2a and D1 nuclei (Supplementary Fig. 3A, B). We hypothesize these differences reflect variability in the number of cells isolated[32], age of mice (6-12 weeks), efficiency of immunoprecipitation[29], differences in DNA amplification or batch effects[33]; rather than biological variation. To address the constraint of global differences in ICuRuS signal (Supplementary Fig. 4A-F), and to identify cell-type specific epigenetic signatures between A2a and D1 nuclei, we assumed ICuRuS signal in A2a and D1 was equal at genes whose expression level was equal between cell-types[34]. Using the publicly available package, ChIPIN[34], we performed inter-sample normalization based on signal invariance across transcriptionally constant genes using cell-type specific Ribo-tag data from A2a and D1 nuclei (Supplementary Fig. 4G, H)[6]. We found H3K27me3

and H3K4me3 ICuRuS signal is equal between A2a and D1 nuclei globally (Fig. 2D, E) and at transcriptionally constant genes (Supplementary Fig. 4G, H).

Next, we directly compared differences in hTPMs at cell-type specific genes using Ribo-tag data from A2a and D1 nuclei[6]. We found that A2a H3K4me3 signal was greater than D1 H3K4me3 signal at genes upregulated in A2a cells and D1 H3K4me3 signal was greater than A2a H3K4me3 signal at genes upregulated in D1 cells (Fig. 3M, N). D1 H3K27me3 signal was greater than A2a H3K27me3 signal at genes upregulated in A2a cells and A2a H3K27me3 signal was greater than D1 H3K27me3 signal at genes upregulated in D1 cells (Fig. 3O, P). Thus, ICuRus generated robust data that facilitated the comparison of chromatin states associated with cell-type specific gene regulation.

## ICuRuS identified H3K4me3/H3K27me3 enriched promotors in A2a and D1

To further our understanding of the combinatorial function of hPTMs in regulating gene expression, we next examined A2a and D1 gene expression as a function of HK4me3/HK27me3 co-enrichment[31] (Fig. 4A, B). Beyond the binary states of active or repressed gene expression, genes co-enriched with H3K4me3 and H3K27me3 show an intermediate level of gene expression mediated by H3K27me3 enrichment[35]. To investigate this concept in specific MSNs, we defined bivalency as coincident H3K4me3 and H3K27me3 enrichment in a window 2 kb upstream and 1 kb downstream of the TSS[30], in A2a and D1 nuclei. ICuRuS found that co-enriched H3K4me3 and H3K27me3 genes are expressed at an intermediate level relative to H3K4me3- or H3K27me3-alone (Wilcoxon rank sum test with Benjamini-Hochberg correction, P < 2e-16; Fig. 4A, B). Next, we calculated the global mutually exclusive index as the random vs real coincidence of two measured hPTMs[36]. We found that the K4me3/K27me3 mutually exclusive index is greater in D1 than A2a nuclei (Fig. 4C), indicating bivalency is less frequently observed in the TSS of D1 MSNs (Fig. 4C). However, A2a, D1 and N2a mutually exclusive index was greater than bulk NAc, which suggests that A2a and D1 ICuRuS revealed distinct features not observed in bulk NAc analysis (Fig. 2C). Most bivalent domains were common to both A2a and D1 MSNs (Fig. 4D) and were enriched for GO biological processes such as negative regulation of cell differentiation, nervous system development and extracellular matrix organization (Fig. 4E).

Next, we sought to identify H3K4me3 and H3K27me3 enrichment patterns associated with stimulus dependent expression of genes that are common or cell-type specific between A2a and D1 MSNs. To identify hPTM profiles associated with stimulus induced gene expression we quantified the co-enrichment of H3K4me3 and H3K27me3 at a previously established set of cocaine-activated genes[37]. The majority of the top ten cocaine activated genes were common between A2a (*Nr4a1*, *Nr4a3*, *Fosb*, *Fosl2*, *Junb*, *Homer1*, *Arc*, *AC134224.3*, *Erg4*, *Penk*) and D1 (*Nr4a1*, *Nr4a3*, *Fosb*, *Fosl2*, *Junb*, *Homer1*, *Arc*, *Tac1*, *Sik2*, *Ntrk2*) and showed H3K27me3 depletion and H3K4me3 enrichment in both A2a and D1 MSN subtypes (Fig. 4F, G, respectively; See Fig. 4H–M for cocaine activated genes in A2a and D1: *Nr4a1*, *Nr4a3*, *Fosl2*; D1 only: *Egr3*; A2a only: *Cartpt*). Most genes were common between A2a and D1 and thus these data failed to provide information regarding the underlying mechanisms of cell-type specific activation. Next, we determined whether H3K4me3 and H3K27me3 enrichment is associated with cell-type specific cocaine activated genes. For example, the immediate early gene, *Egr3*, is activated specifically in D1 neurons by cocaine[23]. Consistent with D1 specific activation, H3K4me3 signal was enriched above background in a window 2 kb upstream and 1 kb downstream of the *Egr3* TSS in both A2a and D1 nuclei. However, H3K27me3 signal was depleted in D1 MSNs relative to A2a MSNs (Fig. 4L; Supplementary Fig. 4A). Whereas *Nr4a1*, *Fosl2*, *Junb*, *Arc*, *Fosb*, *Sik2*, *Ntrk2* show no MSN-subtype specific expression[37] or

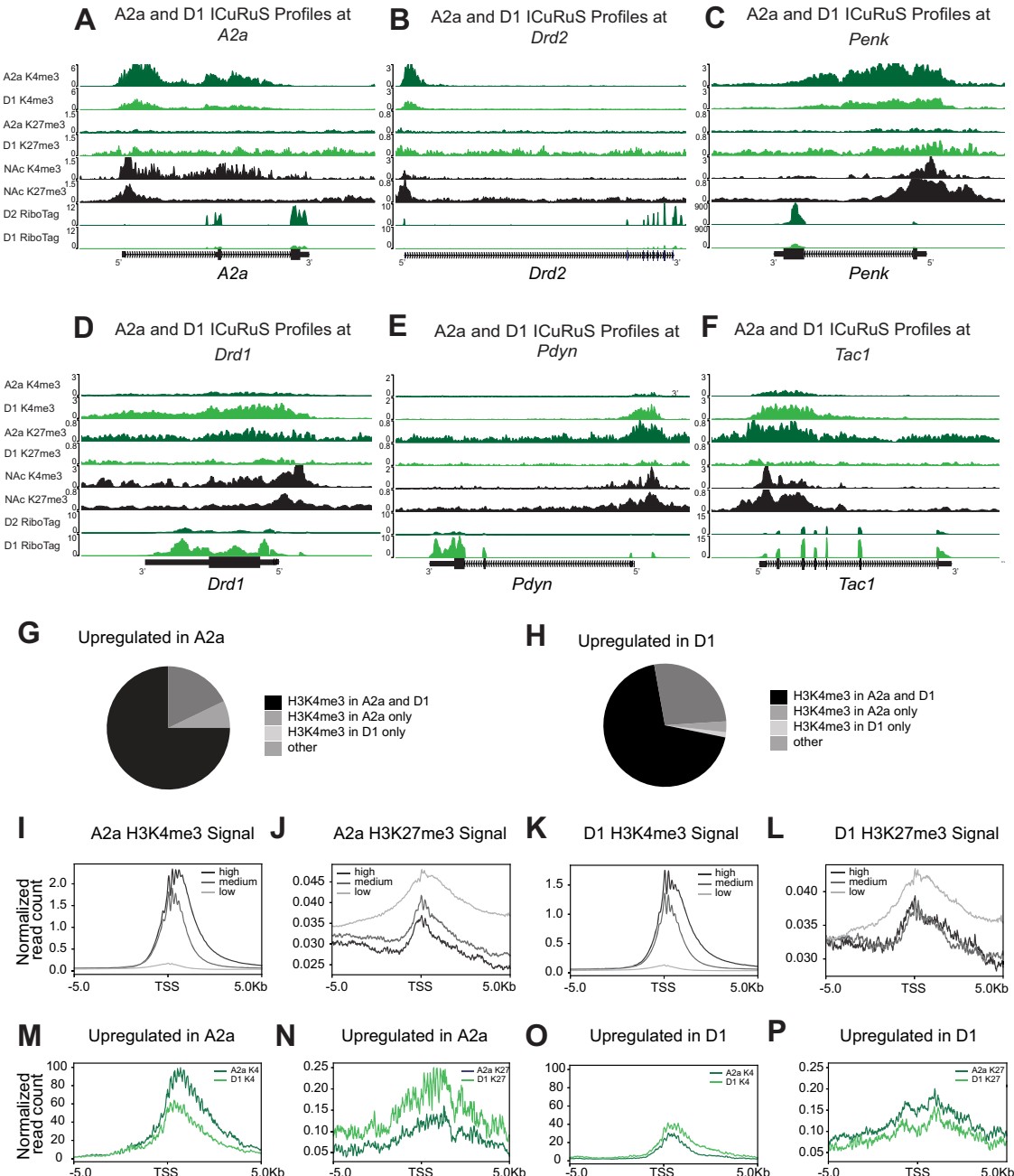

**Fig. 3 | Comparison of peak loci across cell-types. A-C)** A2a and D1 specific H3K4me3 and H3K27me3 modifications around known A2a-specific cell-type markers (**A**) A2a, (B) Drd2, and (**C**) *Penk*. D-F) A2a and D1 specific H3K4me3 and H3K27me3 modifications around known D1-specific cell-type markers (**D**) *Drd1*, (**E**) Pdyn, and (**F**) *Tac1*. (**G**) Peaks at A2a and (**H**) D1 upregulated genes; cell type-specific genes defined by |log2fold change | > 1 and adj *P* < 0.05. (**I**) A2a H3K4me3 signal, (**J**) A2a H3K27me3 signal, (**K**) D1 H3K4me3 signal, (**L**) and D1 H3K27me3 signal centered on transcription start site (TSS). For each cell-type and hPTM, genes are categorized into 3 groups: high (FPKM > = 10), medium (1 < = FPKM < 10), and low

(FPKM < 1) based on published MSN-specific Ribo-Tag[6]. Signal is normalized to total mapped reads: Count Per Million (CPM). **M**) A2a and D1 H3K4me3 signal centered on the TSS of A2a-specific genes, (**N**) A2a and D1 H3K4me3 signal centered on the TSS of D1-specific genes, (**O**) A2a and D1 H3K27me3 signal centered on the TSS of A2a-specific genes, (**P**) A2a and D1 H3K27me3 signal centered on the TSS of D1-specific genes based on published MSN-specific Ribo-Tag[6]. Signal is normalized by variance across transcriptionally constant genes (quantile normalization). A2a and D1-specific genes are defined as *q*-value <0.05 and | log2 FC | > 1.

hPTM enrichment between cell-type, *Egr3*, is differentially activated by cocaine in D1 nuclei[6,38,39] and showed differential H3K27me3 enrichment between A2a or D1 MSNs. Similarly, H3K27me3 is enriched above background at the immediate early gene, *Nr4a2*, in both A2a and D1 nuclei which suggested H3K27me3 is enriched at immediate early genes that escape cocaine-induced activation. Taken together, hPTM profiling by ICuRuS is of sufficient resolution for the analysis of cell-type gene expression in brain.

## Discussion

We found that ICuRuS generated robust and reproducible H3K4me3 and H3K27me3 profiles at sufficient depth to examine cell-type specific chromatin from a single mouse striatum. These findings are an improvement over current methods, such as bulk tissue CnR[25], which obscures cell-type specificity, and single-cell CnR, which requires pooled brain tissue and obscures subject variability[1]. Analysis of a single brain region from a single mouse overcomes a critical barrier

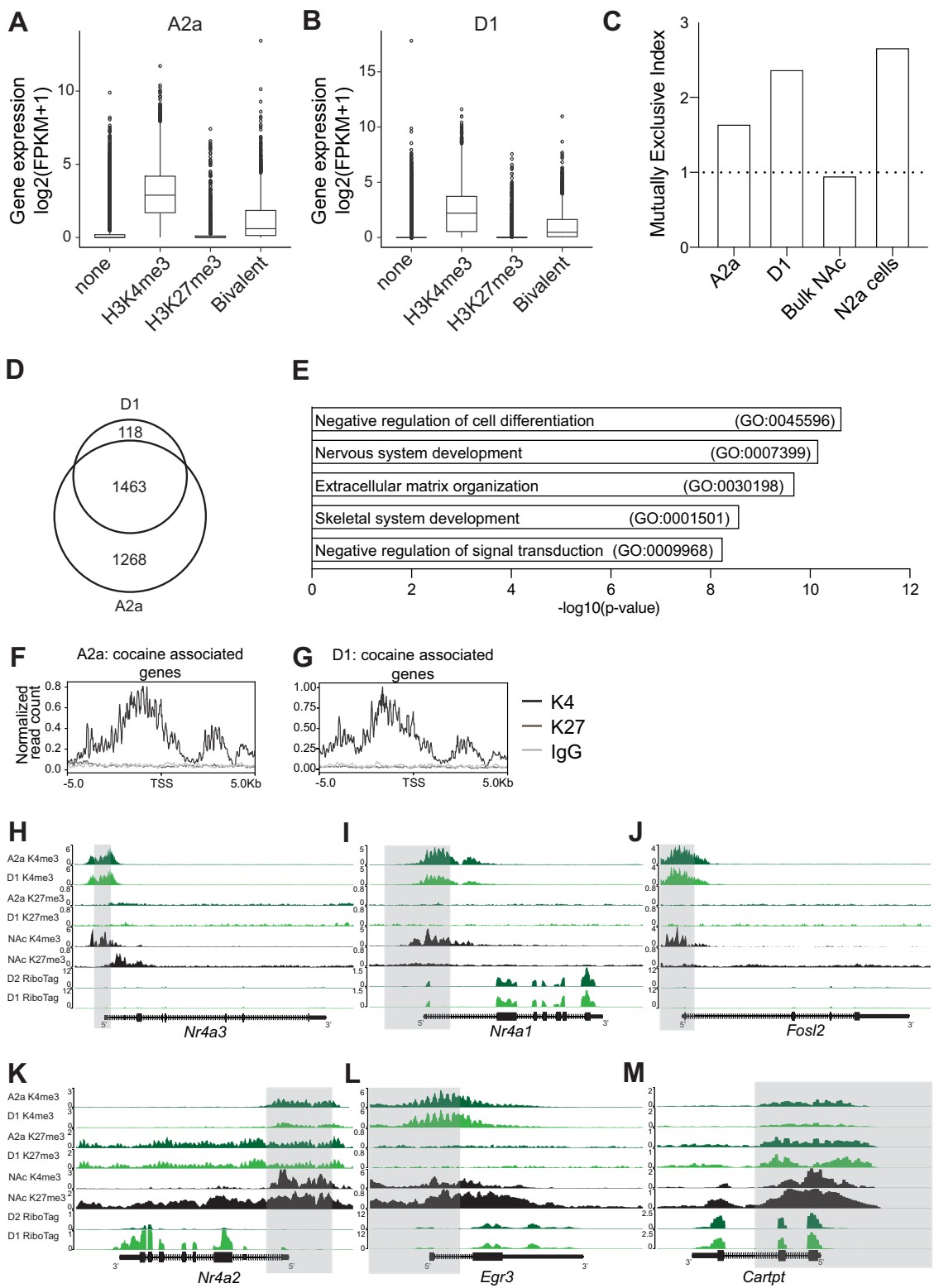

given individual differences in stimulus-induced gene expression and susceptibility to psychiatric disease[40,41]. Recent studies have applied FACS to isolate MSNs, but we found that the affinity purification strategy used in ICuRuS (and in the original INTACT protocol[10]) reduced the probability of false positive signal. In addition, ICuRuS potentially avoids FACS induced ectopic upregulation of activity dependent genes[15], cellular stress[16], artifactual DNA shearing[17], and increased background[18,42]. Finally, ICuRuS allowed identification of H3K4me3/H3K27me3 co-enriched domains at cell-type specific genes,

supporting the hypothesis that neuronal subtype specific gene expression is regulated via combinatorial hPTMs.

To perform ICuRuS we used genetically modified mice and confirmed that Sun1GFP expression in A2a and D1 nuclei had no effects on gross mouse phenotypes. However, subtle physiological effects of the transgene can be avoided by viral delivery of Sun1GFP to single transgenic Cre-expressing mice or rats. To entirely avoid the use of transgenic mice, cell-type specific enhancers or promotors can drive viral Sun1GFP expression[43]. These modifications to ICuRuS would

**Fig. 4 | ICuRuS identified H3K4me3 and H3K27me3 co-enriched promotors in A2a and D1 nuclei. A**) A2a and (**B**) D1 H3K4me3, H3K27me3, and H3K4me3/H3K27me3 combinatorial enrichment between 2 kb upstream and 1 kb downstream of the promoter start site TSS for cell-type specific genes using published MSN-specific Ribo-Tag[6] $n = 2$/cell type and antibody. For D1: Bivalent vs H3K4me3 only: W = 6750380; Bivalent vs H3K27me3 only: W = 7490646. For A2a: Bivalent vs H3K4me3 only: W = 10958639; Bivalent vs H3K27me3 only: W = 18147152. The boxplot is defined by the mean (middle line), interquartile range (lower and upper quartile represent observations outside the 24 –76 percentile range), and unfilled cycles indicating outliers. **C**) H3K4me3/H3K27me3 mutually exclusive index for A2a, D1, N2a, and bulk NAc (bulk NAc raw data was extracted from[28]. A dashed line with an index equal to 1.0 indicates that no exclusive effects exist. **D** Venn diagram displaying cell-type specific and overlapping bivalent domains in A2a and D1 nuclei. **E** GO biological processes for the top five bivalent domains that are present in both A2a and D1 cell-types. **F**) A2a and (**G**) D1 H3K27me3 signal centered on the TSS of cocaine associated genes. Genome browser view of (**H**) *Nr4a3*, (**I**) *Nr4a1*, (**J**) *Fosl2*, (**K**) *Nr4a2*, (**L**) *Egr3*, and (**M**) *Cartpt* for different profiling methods and cell-types. Nucleus accumbens (NAc) bulk ChIP-seq data come from[28]. Ribotag data come from[6]. Signal is normalized to total mapped reads: Count Per Million (CPM) and then by variance across transcriptionally constant genes (quantile normalization).

enable temporal and regional control of Sun1GFP expression. Importantly, CnR can be performed on single cells which would allow for smaller striatal subregions or rare cell-types to be profiled[23,32]. In contrast, ChIP-seq is subject to high background, which is typically overcome by increasing cell input and sequencing depths. We found that H3K4me3 and H3K27me3 profiles defined by CnR and ChIP-seq are largely similar, but fragmentation biases that result in non-uniform coverage in ChIP-seq datasets[44] may affect such comparisons. Consistent fragmentation using MNase may be an important consideration in both ChIP-seq and CnR. Importantly, we found that antibody selection and validation were necessary for optimization of CnR for brain. We screened many antibodies to define those with sufficiently sensitive and specific detection of hPTMs for neuronal epigenome profiling. While ICuRuS can determine H3K4me3/H3K27me3 co-enrichment at specific loci, it cannot determine if these marks occur on the same histone or nucleosome. An alternative approach to study H3K4me3/H3K27me3 bivalency is sequential ChIP, which determines hPTM localization on the same nucleosome[45]. Overall, ICuRuS provides technological advances to understand epigenetic regulation in other neural disorders, brain regions, and cell-types

A2a and D1 MSNs make up ~95% of the neuronal population in the striatum[46]. Cell-type specific gene expression studies have genetically define these subpopulations[6,38], reporting that distinct transcriptional networks in D1-MSNs promote the motivation to seek drugs, while A2a-MSNs generally inhibit this behavior in mice[47]. Conversely, depending on the temporality of stimulation, activation of A2a-MSNs can enhance motivation[48]. Our previous work shows that *Nr4a1* activation in bulk striatal neurons in mouse, reduces the motivation to seek cocaine following abstinence[49]. In whole ventral striatum, Nr4a1 binds and activates downstream target gene, *Cartpt*, via depletion of H3K27me3 and enrichment of H3K4me3 at this locus[49]. Using ICuRuS, we found that H3K27me3 is lowly enriched at *Nr4a1* in A2a and D1 nuclei, consistent with strong activation upon cocaine exposure[49], but neither H3K4me3 nor H3K27me3 are sufficient to specify cell-type specific expression of *Cartpt* and thus other modifications are likely involved. Our lab is adequately equipped to address the causal relevance of these findings using cell-type specific CRISPR activation (e.g. dCas9-VPR) targeted to alternative *Nr4a1* promoters and CRISPR epigenetic editing of H3K27me3 using, Friend of GATA1 (FOG1) and H3K4me3 using mixed lineage leukemia 4 (MLL4), at *Cartpt*. Future studies will use cell-type specific epigenetic editing to determine the exact function of hPTMs at gene loci and how cell-type specific hPTM enrichment changes in response to stimuli.

Traditionally, H3K4me3 and H3K27me3 co-enriched genes have been described as 'poised' – silenced in neuronal precursor cells but expressed in a cell-type specific manner following differentiation[31]. Consistent with the literature, MSN ICuRuS profiling found H3K4me3 and H3K27me3 were independently enriched at expressed and repressed genes, respectively[6]. However, genes co-enriched in H3K4me3 and H3K27me3 showed intermediate levels of gene expression across both MSN subtypes. These data suggest competitive antagonism between the transcription factors and RNA polymerase recruited by H3K4me3 and H3K27me3[36], such that cell-type specific

expression of transcription factors may explain discordance in binary gene regulation[50]. In fact, the activator CREB is bound at H3K27me3 depleted regions in response to neural activity[51]. Cocaine-induced phosphorylation of CREB[52] bound to H3K27me3 depleted promotors is an attractive mechanism for stimulus induced cell-type specific gene activation.

While consideration of combinatorial hPTM function is crucial to interpreting the histone code, current data is sparse on the mechanism underlying cell-type specific and activity dependent gene expression in the adult mammalian brain. Cocaine-activated genes, *Homer1, Nr4a3, Nr4a1, Fosl2, Junb, Tac1, Arc, Fosb, Sik2,* and *Ntrk2*, lack broad H3K27me3 domains in A2a and D1 nuclei[53], suggesting that depletion of repressive hPTMs is involved in the activation of these genes. Whereas *Nr4a1, Nr4a3, Fosl2, Junb, Arc, Fosb, Sik2, Ntrk2* show no MSN-subtype specific expression or hPTM enrichment, *Cartpt, Neurod6, Oprk1* were differentially expressed between D1 and A2a cells[6,38–40,54] and showed differential H3K27me3 and H3K4me3 enrichment in either A2a or D1. For example, D1-specific expression of *Neurod6* was associated with depletion of H3K4me3 in D1 MSNs relative A2a MSNs[6,39], and A2a specific expression of *Oprk1* was associated with depletion of H3K27me3 in A2a MSNs[6,39]. In this way, neuronal subtype specific expression is conferred by either loss of permissive hPTMs in the non-expressing subtype, or loss of a repressive hPTM in the expressing subtype. Alternatively, D1-specific *Bdnf* and *Nr4a2* expression were not associated with cell-type specific differences in H3K4me3 or H3K27me3 enrichment, suggesting additional modes of epigenetic regulation. However, some genes did show differential hPTM enrichment patterns that were associated with cell-type specific activation following cocaine exposure. For example, cocaine induced A2a-specific *Penk* expression is associated with depletion of H3K27me3 and enrichment of H3K4me3 in A2a nuclei relative to D1 nuclei. Cocaine induced D1-specific *Tac1* expression is associated with depletion of H3K27me3 and enrichment of H3K4me3 in D1 nuclei relative to A2a nuclei. In addition, D1-specific *Egr3* cocaine induced activation is associated with depletion of H3K27me3 in D1 nuclei relative to A2a nuclei despite similar levels of H3K4me3. Taken together, we hypothesize de-repression is a possible mechanism for some MSN-specific expression patterns.

ICuRuS is a powerful tool for investigations of cell-type specific epigenetic gene regulation in brain. First, it provides high resolution hPTM profiles from low cell numbers recovered from a single brain region of a single mouse. We obtained good quality hPTM profiling data with cell type specific resolution. In this way, ICuRuS can be paired with cell-type specific epigenetic editing tools to establish causal roles of hPTMs in the regulation of gene expression and behavior. Altogether, our findings complement and expand upon existing methods for the examination of cell-type specific global chromatin changes in brain.

## Methods

### Animals

The R26-CAG-LSL-Sun1-sfGFP knock-in mouse on the C57BL/6 J background was crossed with A2a-cre and *Drd1*-cre mice on a C57BL/6 J background to generate Sun1-sfGFP;A2a-cre and Sun1-sfGFP;Drd1-cre

mouse lines. All mice were purchased from Jackson Laboratory. Mice were housed on a 12-h light-dark cycle at constant temperature (23 °C) and humidity (40-60%) with access to food (Laboratory Rodent Diet 5001) and water ad libitum. All animal procedures were conducted in accordance with the National Institutes of Health Guidelines as well as the Association for Assessment and Accreditation of Laboratory Animal Care. Ethical and experimental considerations were approved by the Institutional Animal Care and Use Committee of The University of Pennsylvania (Protocol # 805959). Male and Female mice at ~8 weeks of age were allocated equally into groups and sex was not considered in the study design.

## INTACT

For each experiment, bilateral striatum of mouse was rapidly dissected in ice-cold homogenization buffer (0.25 M sucrose, 25 mM KCl, 5 mM MgCl$_2$, 20 mM Tricine-NaOH). The tissue was the slow frozen at −80 °C until INTACT was conducted. The INTACT procedure used was modified from[10]. To initiate the INTACT procedure, tissue was Dounce homogenized using a loose pestle (10 strokes) in 1.2 mL of homogenization buffer supplemented with 1 mM DTT, 0.15 mM spermine, 0.5 mM spermidine, and EDTA-free protease inhibitor (Roche). A 5% IGEPAL-630 solution was added, and the homogenate was further homogenized with the tight pestle (7 strokes). The sample was then mixed with 1.3 mL of 50% iodixanol density medium (Sigma D1556), and added to an Ultra-Clear Tube (13.2 mL, Beckman and Coulter, 344059). The sample was then underlayed with a gradient of 30% and 40% iodixanol, and ultra-centrifuged at 28900 x g for 18 minutes in a swinging bucket centrifuge at 4 °C. Nuclei were collected at the 30%-40% interface and pre-cleared by incubating with 20 μL of Protein G Dynabeads (Life Technologies 10003D) for 15 minutes. After removing the beads with a magnet, the mixture was incubated with 10 μL of 0.2 mg/mL rabbit monoclonal anti-GFP antibody (Life Technologies G10362) for 30 minutes. 60 μL of Dynabeads were added, and the mixture was incubated for an additional 20 minutes. To increase yield, the bead-nuclei mixture was placed on a magnet for 30 seconds to 1 minute, completely resuspended by inversion, and placed back on the magnet. This was repeated seven times. Bead-bound nuclei were then washed 3 ×800 uL with wash buffer. All steps were performed on ice or in the cold room, and all incubations were carried out using an end-to-end rotator. When purifying RNA following INTACT, RNasin ® Plus Ribonuclease Inhibitor (Promega N2611) was added to the antibody and bead buffers. Quantitative polymerase chain reaction (qPCR) was performed using standard methods. Primer for qPCR: A2A (Fig. 1F, G) F: ACTCTCCCCTCCACACCC R: CATAGTTTCTGTCTTCCAGCCC; *Drd1* (Fig. 1F, G) F: TTCTTCCTGGTATGGCTTGG R: GCTTAGCCCT-CACGTTCTTG; Cartpt (Supplementary Fig. 5B) F: ACGA-GAAGGAGCTGATCGAA R: TCTCTGAGGGGAACGCAAAC.

## Cell counting

For cell counting to quantify INTACT specificity and yield, during the INTACT procedure aliquots were collected following nuclei isolation and following affinity bead-bound pull downs. Nuclei from isolation (i.e. before the addition of antibody and Protein G beads) and bead-bound nuclei were stained with DAPI (30uM) and 10ul of sample was added to a hemocytometer. Images were acquired using a Leica Fluorescence Microscope. DAPI + /GFP + and DAPI + /GFP- nuclei were counted using a hemocytometer. The following equations[10] were used to calculate the specificity (A) and yield (B) from INTACT:

$$(A) \frac{(DAPI+/GFP+)}{(DAPI+/GFP+) + (DAPI+/GFP-)} \text{ x aliquot scale}$$

$$(B) \frac{(Bead-bound\ DAPI+/GFP+)}{(Nuclei\ Isolation\ DAPI+/GFP+)} \text{ x aliquot scale.}$$

(1)

## CnR (following INTACT)

As the INTACT protocol[12,13] relies on bead-immunopurification of Sun1-GFP + nuclei, it is ideally suited to downstream processing of bead-immobilized nuclei by CnR. CnR was carried out according to published protocols[13], with changes pertinent to neuronal isolation. Nuclei from one striatum (~10,000) were washed 2X and then resuspended in Digitonin Buffer with either anti-H3K4me3 (Antibody 1: 1:50 dilution, Abcam, Ab8580 (Fig. 2B, D); Antibody 2: 1:50 dilution, Active Motif, 39159 (Supplementary Fig. 2); Antibody 3: 1:50 dilution, Epicyphr, 13-0041(Supplementary Fig. 2), anti-H3K27me3 (Antibody 1: 1:100 dilution, Active Motif, 39055 (Fig. 2C, E); Antibody 2: 1:100 dilution, Thermo Fisher, MA5-1198 (Supplementary Fig. 2) or IgG (1:100; Epicypher, 13-0042 (Fig. 2 and Supplementary Fig. 2) for two hours, 4 °C with rotation. Following incubation, bead-bound nuclei were washed 2X with Digitonin Buffer, and antibody-enriched fragments were cut by one-hour, 4 °C incubation with 0.5 ul MNAse (20x CUTANA™ pAG-MNase 15-1116). Following MNAse incubation, bead-bound nuclei were washed 2X with Digitonin Buffer, and antibody-enriched fragments were released by two-hour, 4 °C incubation with 100 mM CaCl2. Antibody-enriched fragments from bead-bound nuclei were then washed again and the reaction was stopped with 33 μl Stop Buffer (68 ul of 5 M NaCl, 40 μl of 0.5 M EDTA, 40 μl of 4 mM EGTA) and incubated for 10 min (37 °C). 1 μl of 10% (wt/vol) SDS and 1.5 μl of proteinase K (20 mg/ml) were added to each sample, followed by 20 min (50 °C) incubation. Total DNA from each sample was purified using phenol/chloroform/isoamyl alcohol and dissolved in TE.

## Library prep

Library preparation of CnR was performed using a NEB Ultra II library preparation kit, with modifications to the protocol[24]. CnR DNA ( < 5 ng) was used as input. End preparation was performed for 30 minutes at 20 °C followed by 1 hour at 50 °C. Adapter at .6 pmol was ligated to end preparation products at 20 °C for 15 minutes. USER enzyme was added, and DNA was purified using 1.1X AMPure beads. Libraries were amplified with 2x Ultra Q5 min, universal primer, and index primers. PCR was as follows: 98 °C for 20 s; 2 cycles of 98 °C for 10 s and 65 °C for 10 s; and a final extension at 65 °C for 5 min. PCR products were size selected using 1.1X AMPure beads. PCR products were quantified using quBIT, Agilent Bioanalyzer and NEB library quant kit (E7630L). Libraries were pooled at equimolar amounts and sequenced using the Nextseq500 platform. Paired-end sequencing was performed (read length, 38 bp × 2; index, 8 bp).

## Immunohistochemistry

Sun1-GFP; A2a-cre and Sun1-GFP; Drd1-cre mice were anesthetized with ketamine/xylazine and perfused with 4% paraformaldehyde (PFA). Brains were then rapidly extracted and stored in 4% PFA overnight for fixation. Brains were then transferred to 15% sucrose for 24 hours and then stored in 30% sucrose until sectioning (40 μm) using a cryostat. Sections were blocked for 90 minutes at room temperature with 10% Normal Donkey Serum and 0.2% Triton X-100 in PBS and then incubated with 0.2% Triton X-100 in PBS with the following antibodies overnight at 4 °C: rabbit anti-Drd1 (1:500, Bioss USA BSM-52920R) or rabbit anti-A2a (1:200, Fisher Scientific PA1-042), co-incubated with goat anti-GFP (1:500, Rockland, 600-101-215). Sections were then washed three times with PBS and incubated at room temperature in the dark for two hours with secondary antibodies for fluorescent labeling: Donkey anti-Rabbit IgG (H + L) Highly Cross-Adsorbed Secondary Antibody, Alexa Fluor Plus 555 (Fisher Scientific A32794, 1:1000 dilution) and Donkey anti-Goat IgG (H + L) Cross-Adsorbed Secondary Antibody (1:1000 dilution), Alexa Fluor 488 (Fisher Scientific A-11055 1:1000 dilution). After the first hour of incubation, 0.5 μg/mL of DAPI (dihydrochloride, Invitrogen D1306) was directly added into the secondary antibody solution. Sections were then washed three times with

PBS and mounted on glass slides using ProLong™ Gold Antifade Mountant (Invitrogen P36930). Images were taken using a Zeiss LSM 810 Confocal Microscope.

## Published ChIP-seq data analysis

N2a and nucleus accumbens (NAc) ChIP-seq data were previously published (H3K4me3[26]; H3K27me3[27]; NAc H3K4me3 and H3K27me3[28]). Raw reads were aligned to the mm10 reference genome using bowtie2[55] with parameters: -q --no-unal --phred33. Uniquely mapped reads (mapping quality score >= 20) were selected using samtools with command samtools view -q 20 (version 0.1.19)[56].

## CnR data processing and analysis

Base call (BCL) files were demultiplexed and converted into FASTQ files using bcl2fastq2 (version v2.20.0.422) with default parameters. Next, raw paired-end reads were mapped to the mm10 reference genome using bowtie2 (version 2.1.0)[55] with options: -q --local --no-mixed --no-unal --dovetail --phred33. Using local alignment mode makes removal of adapter sequences unnecessary. Uniquely mapped reads (mapping quality score >= 20) were selected using samtools (version 0.1.19, samtools view -q 20). Then, we used Picard (version: 2.23.4)[57] to check whether the insert size distribution is consistent with the library fragment size obtained from Bioanaylzer. Next, duplicates were removed using Picard MarksDuplicates function. We selected deduplicated mapped reads with insert size between 150 and 500 bp using samtools for downstream analyses. Read alignments were normalized to total mapped reads using deepTools with command: bamCoverage –bam inbam -o outbw –normalization. D1 and A2a H3K4me3 and H3K27me3 signals were further quantile normalized using the CHIPIN[34] with the expression data from RiboTag-purified Drd1 and Drd2 neurons of the NAc[6]. Tracks were visualized by deepTools pyGenomeTracks (https://github.com/deeptools/pyGenomeTracks).

## RNA-seq data processing and analysis

Drd1 and A2a cell-type-specific RNA-seq data were derived from RiboTag-purified Drd1 and Drd2 neurons of the NAc, respectively[6]. Raw reads were mapped to the mm10 reference genome and Gencode annotation (vM23) using STAR[58] with parameters: --outFilterMismatchNmax 3 --outFilterMultimapNmax 1 --alignSJoverhangMin 8. Differentially expressed genes between Drd1 and Drd2 neurons were detected using cufflinks cuffdiff with default parameters (v2.2.0)[59]. Read alignments were normalized to total mapped reads using deepTools for visualization.

## Correlation between ICuRuS and ChIP-seq data

To evaluate reproducibility and specificity of the ICuRuS data, we calculated pairwise Pearson's correlation coefficients among ICuRuS replicates and corresponding ChIP-seq datasets. We first computed with read coverages for the entire genome using deepTools with multiBamSummary function. Then, correlation coefficients were calculated and visualized by plotCorrelation function of deepTools[60] with parameters: --skipZeros.

## Signal around peaks or TSS

For signal around H3K4me3 peaks and TSS (+/− 5 kb), we calculated signal using deepTools with command: computeMatrix reference-point -a 5000 -b 5000. For signal around H3K27me3 peaks, peaks 5 kb apart were first merged using the bedtools merge[61] with parameter: -d 5000. Then, H3K27me3 signal were calculated using deepTools with command: computeMatrix scale-regions -a 5000 -b 5000. Signal around each peak or TSS and was plotted using deepTools plotHeatmap function. Averaged signal was plotted using deepTools plotProfile function.

## Peak calling

H3K4me3 peaks were called using MACS2[62] with parameters: -f BAM -g mm -B --keep-dup all -q 0.01. H3K27me3 peaks were called using SICER[63] with the command: sicer -t inbam -c control -s mm10. H3K27me3 peaks with false discovery rate less than 0.01 were selected. Overlapped peaks were counted using HOMER mergePeaks function[64] and visualized by Upset plots[65]. Fraction of reads in peaks was calculated using a custom Python script.

## Bivalency analysis

We categorized genes into four groups based on whether there are H3K4me3 and/or H3K27me3 peaks on promotor regions (−2 kb to +1 kb)[30]: 1) Bivalent genes which have both H3K4me3 and H3K27me3 peaks, 2) H3K4me3 genes which have only H3K4me3 peaks, 3) H3K27me3 genes which have only H3K27me3 peaks, and 4) none genes which have no peaks. Mutually exclusive index was calculated based on a previous method [71].

## Statistics & reproducibility

The data were analyzed using GraphPad Prism software. The results are reported as the means ± SEMs or mean ± SDs of at least two biological replicates for each experiment. Mice were randomly allocated to groups. The investigators were not blinded to allocation of mice during experiments because genotypes were confirmed visually prior to dissection but investigators were blinded during the computational outcome assessment. The significance of differences between two groups was assessed by Student's t test, the Mann-Whitney U-test and the Wilcoxon signed-rank test. No statistical method was used to predetermine sample size. No data were excluded from the analyses. Statistical significance was defined as follow: *p value <0.05, **p value <0.01 or ***p value <0.001.

## Reporting summary

Further information on research design is available in the Nature Portfolio Reporting Summary linked to this article.

# Data availability

All raw and processed sequencing data generated in this study have been deposited to NCBI Gene Expression Omnibus under accession number: GSE193673. The qPCR and cell counting data generated in this study are provided in the Supplementary Information/Source Data file. Data used in these studies have been deposited. N2a H3K4me3 ChIP-seq: GSE91043; N2a H3K27me3 ChIP-seq: GSE107310; NAc H3K4me3 and H3K27me3 ChIP-seq: GSE42811, NAc RNA-seq: GSE121199. Upon request to the corresponding author, all raw data is available and will be provided within two weeks from the request date. Source data are provided with this paper.

# Code availability

All scripts are available at github (https://github.com/HellerLAbeats/ICuRuS).

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

## Acknowledgements

Financial support was provided by NIH-NIDA Avenir Director's Pioneer Award (E.A.H, DP1 DA044250), NIDA Research Project Grant (E.A.H, 1R01DA052465-01A1), Pilot Award from the Epigenetics Institute at the University of Pennsylvania, NIH Drug Abuse Dissertation Research Award (M.D.C. R36 DA050877), Children's Hospital of Philadelphia Post-doctoral Fellowship for Academic Diversity (M.D.C), T32 Predoctoral Training Grant in Addiction (D.K.F. NIDA T32DA028874), and SynGAP Research Fund Postdoctoral Fellowship (S.Z.). We thank Dr. Yemin Lan for independently validating some findings present in this study. We thank the Next Generation Sequencing Core at University of Pennsylvania for consultation on this project. Figure 1C and Supplementary Fig. 2A were created with Biorender.com.

## Author contributions

M.D.C., D.K.F. and S.Z contributed equally to this work. M.D.C., D.K.F. and E.A.H designed and executed the in vivo ICuRuS experiments. M.D.C., and E.A.H designed and executed the in vitro CUT&RUN experiments. D.K.F and E.A.H designed and analyzed the qPCR experiments. M.D.C., S.Z and E.A.H developed algorithms and analyzed the sequencing data. D.K.F., A.M.B., H.S., and E.A.H. designed and analyzed IHC experiments. K.S.C and M.T.W assisted in animal husbandry. D.K.F and E.A.H. designed the schematics. M.D.C. and E.A.H. wrote the first draft of the manuscript. M.D.C., D.K.F., S.Z. and E.A.H. revised the manuscript. M.D.C., D.K.F., S.Z., A.M.B., K.S.C., M.T.W, H.S. and E.A.H. approved the final version of the manuscript.

## Competing interests

The authors declare no competing interests.
