## [Peer Review File · Nature Communications]

Reviewers' Comments:

Reviewer #1:

Remarks to the Author:

Summary

Carpenter et al develop an assay to efficiently measure histone modifications in labeled populations of cells by combining INTACT, CUT&RUN-Seq, and ChIP-Seq into an experimental procedure called ICuRuS. They apply this technique to measure H3K4me3 and H3K27me3 in sub-populations of medium spiny neurons. This new experimental technique has the potential to be applied much more broadly and is of interest to the community. The data that is collected in the manuscript is quite high quality. Browser plots of marker genes show a reliable (often cell type-specific) signal and the analysis on the individual samples passes QC metrics. However, the manuscript could be strengthened with more direct comparisons of the epigenetic data to show the extent to which these datasets are cell type specific. For example, it is important that differential peaks/regions be called between D1 and D2 MSNs and that those differences are compared to cell type-specific gene expression signatures. In addition, the comparison of activity-dependent genes to co-occurring H3K4me3/H3K27me3 would be stronger if a more systematic comparison were conducted that demonstrated an association between cell type-specific activation and cell type-specific histone mark co-occurrence.

Strengths

This technology makes it feasible to perform cell type-specific profiling of histone modifications using transgenic mice. Increasingly, research in both molecular and neural circuit features of complex traits are uncovering mechanisms that are highly cell type specific. Although the process of addiction is highlighted in this manuscript, ICuRuS has the potential to be applied to a number of other neural disorders, brain regions, and cell types.

There is substantial quality control evidence that clean epigenetic data is being acquired.

Major Issues

The results in Figure 3 and 4 are encouraging, demonstrating that within each cell type gene expression is positively associated with H3K4me3 and negatively associated with H3K27me3. However, to show that the method can identify cell type-specific signals, it is necessary to compare the samples D1 and D2 samples more directly. Using the presence or absence of a peak in one cell type versus another can lead to false positives. For example, a peak could be just below the threshold in one cell type and just above threshold in the other. Methods that directly compare read counts or signal across peaks should be applied to identify regions with a bias for D1 and D2 MSNs. Those differential regions should be associated with differential gene expression Ribo-Tag.

The "poised" chromatin state has traditionally referred to more developmentally poised as opposed to poised for neural activity-dependent stimulation. For example, Kim et al 2010 (<https://pubmed.ncbi.nlm.nih.gov/20393465/>) notice a depletion of H3K27me3 at enhancers that bind CBP in response to neural activity. Is there additional evidence to suggest that bivalent/co-enriched regions should be stimulus-dependent? Within this dataset are known IEGs or cell type-specific IEGs enriched for co-occurring H3K27me3/H3K4me3?

Minor Issues

- Remove "149 wds" from abstract header
- The "In contrast" header doesn't seem to make sense, since both pieces of evidence support the role of MSNs in disease processes. (line 53)
- Reference 23 (Skene and Henikoff) does not seem to address the increase sequencing noise with FACS. In addition, identifying a cell type-specific nuclear protein and the right antibody are other major limitations of FACS.
- The methods mention "specificity" and yield, but Fig. 1 uses the terminology "purity" and yield.
- Fig1F and Fig1G – The statistical test applied and the p-values for the comparisons are not

reported.

- Fig2D – The complete absence of any signal (or even noise) from the input is hard to believe. If true, why are the input values so different for the two antibodies
- Fig3 – It would be helpful if the direction arrows on the transcripts were larger so that the 5' and 3' ends of the genes can be easily identified.

Reviewer #2:

Remarks to the Author:

The study by Carpenter et al. combined cell-type specific nuclei isolation by the INTACT method with CUT&RUN in order to profile active (H3K4me3) and repressive (H3K27me3) histone modifications in two medium spiny neuron (MSN) subtypes of the mouse striatum, namely Adenosine 2a Receptor (A2a)- expressing and Dopamine Receptor D1 (D1)-expressing neurons. The authors first test the efficiency of the H3K4me3 and H3K27me3 CUT&RUN using the N2a cell line by comparing the CUT&RUN enrichment data to the available corresponding ChIP-seq data from the same cell line. Once the CUT&RUN protocol was established, they work with the INTACT-isolated A2a and D1 nuclei from the mouse brain and show comparable H3K4me3 and H3K27me3 enrichment data to the available corresponding ChIP-seq data from the bulk mouse nucleus accumbens brain tissue. The authors conclude that their INTACT-CUT&RUN protocol allows profiling of both active and repressive histone marks in a cell-type specific way in a brain tissue obtained from a single mouse, using only around 10,000 neuronal nuclei, which would not be permissive with current ChIP-seq protocols. They also highlight the advantages of not using FACS sorting (for cell type purification) or formaldehyde fixation (for histone modification profiling) in their current protocol as these procedures may lead to artifacts and increased background.

I believe that this is a very valuable tool for neuroscientists and would like to applaud the authors for their comprehensive protocol optimization and comparisons, including not only using both cell line and brain tissue but also multiple antibodies to test the efficiency of their protocol.

However, while this protocol is valuable, I believe that, by itself, this study does not offer any novelty in terms of biological insight into gene regulation in the brain and this is its major limitation. I would suggest that the authors expand on this study and use the developed protocol to answer a biological question regarding cell type-specific chromatin and gene regulation in striatal MSNs. In addition, I would like to offer a couple of comments that may be useful to the authors.

1) I agree that both FACS and formaldehyde crosslinking have their own limitations. However, the INTACT method requires using genetically modified animals. While these mice may not show gross changes in phenotype, subtle changes in mouse physiology may compromise some studies – I believe that the authors should discuss this. In addition, CUT&RUN may be efficient for histone modifications but less so for transcription factor binding (or this may require cross-linking too?). In general, native ChIP also does not require cross-linking and is a very efficient way of profiling histone modifications. More generally, the authors should discuss, in more detail, both the advantages and disadvantages of their approach compared to ChIP-seq, which is still used as the standard histone modification profiling technique by many labs, in a more naturalistic setting.

2) Since this is a very nice protocol that could be helpful to many research groups and the method development is the main contribution of this manuscript, I suggest that the authors provide a step-by-step protocol as part of their Supplementary Information. In addition, it would be nice to provide experimental quality control (QC) checkpoints that are used in the protocol to confirm that the procedure proceeds well such as bioanalyzer library prep traces, Qubit measurements, etc. Having a step-by-step protocol would likely allow for broader use of this method by other research groups.

Reviewer #3:

Remarks to the Author:

This manuscript by Carpenter, Fischer, Zhang, et al describes a new protocol using INTACT and CUT&RUN (CnR) methods, together ICuRUS, to profile cell-type specific histone modifications in individual mouse brain tissue samples. Using these combined methods overcomes the limitations of other neuro-epigenomic techniques such as ChIP-Seq, which cannot perform cell-type specific analysis or incorporate information on the individual animal's behavior or disease pathology due to the required high amount of starting material (i.e. pooling of samples). This method was then applied to profile 2 histone methylation marks, a marker of transcriptional activation (H3K4me3) and a marker of transcriptional repression (H3K27me3) within 2 striatal cell types, D1R-expressing and A2a-expressing cells, to determine how these marks regulate cell-type specific gene expression.

The authors first demonstrated this protocol's ability to successfully isolate cell-type specific nuclei from the striatum of either D1R-Cre; or A2a-Cre; SUN1-GFP transgenic mice using RT-qPCR and immunohistochemistry. Separately, their CnR method using N2a cells produced high quality reads and similar distributions of both H3K4me3 and H3K27me3 reads in comparison to previously published N2a ChIP-Seq datasets. Following the validation of both separate approaches, the authors used ICuRuS to profile H3K4me3 and H3K27me3 in D1R- and A2a- nuclei from mouse striatal tissue. ICuRuS replicate datasets were highly reproducible, and overall displayed consistent pattern of read coverage to ChIP-Seq. To compare the relationship between cell-type specific histone marks and mRNA expression, the authors next overlaid their ICuRuS datasets with previously published D1R and D2R-specific transcriptome datasets. As expected, cell-type specific H3K4me3 reads were enriched in genes highly expressed in both D1R and A2a as H3K27me3 were enriched in lowly expressed genes. Finally, the combinatorial function of H3K4me3 and H3K27me3 were investigated to determine whether a subset of genes enriched for both marks were poised for high expression upon stimulation. Potential higher sensitivity for detecting co-enrichment was seen from CnR and ICuRuS datasets versus ChIP-Seq. Majority of co-enriched domains from ICuRuS data were seen in both A2a and D1 MSNs, and these genes were associated with processes relating to cell differentiation, development. However, immediate early genes (IEGs) did not display a consistent pattern of co-enrichment and genes expressed within a specific cell-types did not display a co-enrichment pattern specific to a cell-types.

Overall, the authors present a thorough validation of their protocol which optimizes the combined use of 2 previously established methods for cell-type specific epigenetic profiling. Their use of previously published datasets to analyze and confirm the accuracy of this protocol is also well-performed. However, this analysis produces limited new conclusions for the field. For instance, their results confirm that well established repressive and permissive histone marks correlate with low and high gene expression respectively in these striatal cell types at baseline. When examining the potential role of co-enrichment of these histone marks on gene expression to poise gene expression, majority of the co-enriched domains were common in both cell types, indicating a lack of genes that are poised for expression in a cell-type specific manner via these histone marks. Further, when investigating co-enrichment of histone marks on immediate early genes, which the authors suggested would likely be poised in either cell type for responses to stimulation, no pattern was observed. This manuscript presents a useful new protocol for cell-type specific epigenetic profiling, however we provide suggestions below to enhance its significance to field.

1) It is appreciated that the authors show mRNA enrichment or depletion of A2A and D1 in Figure 1 and H3K4me3 or H3K27me3 enrichment on A2A or D1 promoters in Figure 3. However, the study would benefit from further cell type specific validation including enrichment of H3K4me3 or H3K27me3 on promoters of other known enriched genes (i.e. A2A MSNs- D2, Penk, Gpr6; D1 MSNs- Pdyn, Tac1, Chrm4). This would be especially important since the enrichment in A2A and D2 promoters may include the BAC transgene which uses these promoters.

2) To use this method to further investigate the role of combinatorial histone marks on poising gene expression, the authors could perform additional experiments to identify whether these combined marks or others are critical regulators. Considering the expertise of the authors, this could be performed by examining whether chronic cocaine exposure impacts the co-enrichment of these histone marks on immediate early genes within D1 or A2A striatal cell types. Cocaine exposure can induce differential effects on IEG expression in these cell types, therefore this treatment may better parse apart whether these combined marks poise future gene expression

based on prior experience. In addition, the effects of cocaine on cell-type specific histone modifications would yield highly useful data for the field as this could be overlaid with cell-type specific RNA-Seq data sets.

3) The authors also compare their protocol to other isolation methods such as FACS and emphasize how their technique prevents false signals. Direct comparisons of those approaches would further demonstrate the importance of their protocol and be critical information to the field.

4) Many studies identify distinct behavioral outcomes through manipulating MSN subtypes across striatal regions (i.e. dorsal lateral striatum, dorsal medial striatum, and different regions of the nucleus accumbens). Presumably these regions have distinct epigenetic/transcriptome adaptations that correlate with behavioral outcomes. The current study isolates total striatum for cell type specific analysis. Do the authors think they can perform the histone post translational modification profiles in these striatal regions or are they still limited by material especially in smaller nucleus accumbens regions? The authors might include some discussion on this in the discussion section.

Reviewer #1 (Remarks to the Author):

Summary

Carpenter et al develop an assay to efficiently measure histone modifications in labeled populations of cells by combining INTACT, CUT&RUN-Seq, and ChIP-Seq into an experimental procedure called ICuRuS. They apply this technique to measure H3K4me3 and H3K27me3 in sub-populations of medium spiny neurons. This new experimental technique has the potential to be applied much more broadly and is of interest to the community. The data that is collected in the manuscript is quite high quality. Browser plots of marker genes show a reliable (often cell type-specific) signal and the analysis on the individual samples passes QC metrics.

We appreciate the reviewers' comments that ICuRuS generates high quality profiling data and has the potential to be applied broadly to other neural disorders, brain regions, and cell types.

However, the manuscript could be strengthened with more direct comparisons of the epigenetic data to show the extent to which these datasets are cell type specific. For example, it is important that differential peaks/regions be called between D1 and D2 MSNs and that those differences are compared to cell type-specific gene expression signatures.

The revised manuscript includes new data and analyses that directly compares A2a and D1 nuclei and quantifies cell type specific gene expression in brain (See: Results, Page 9, Lines 211-225). We applied a novel normalization technique to establish a baseline across samples that can be used to identify differential peaks/regions between A2a and D1 MSNs (See: Results, Page 8, Lines 183-194). In addition, we examined hPTMs at genes activated by cocaine in A2a or D1 MSNs, and found these genes are depleted in H3K27me3 (See: Results, Page 10, Lines 245-254).

In addition, the comparison of activity-dependent genes to co-occurring H3K4me3/H3K27me3 would be stronger if a more systematic comparison were conducted that demonstrated an association between cell type-specific activation and cell type-specific histone mark co-occurrence.

We thank the reviewer for pushing us to ask more in-depth questions using our data including systematic comparisons to demonstrate the association between cell type specific activation and hPTM enrichment. We include new data that demonstrates cell type-specific mRNA level (See: Results, Page 9, Lines 211-225) and cell type-specific histone mark co-occurrence (See: Results, Page 10, Lines 257-266). Overall, we provide new conclusions regarding A2a- and D1-specific H3K27me3 enrichment and its importance in activity dependent transcription.

Strengths

This technology makes it feasible to perform cell type-specific profiling of histone modifications using transgenic mice. Increasingly, research in both molecular and neural circuit features of complex traits are uncovering mechanisms that are highly cell type specific. Although the process of addiction is highlighted in this manuscript, ICuRuS has the potential to be applied to a number of other neural disorders, brain regions, and cell types. There is substantial quality control evidence that clean epigenetic data is being acquired.

Major Issues

The results in Figure 3 and 4 are encouraging, demonstrating that within each cell type gene expression is positively associated with H3K4me3 and negatively associated with H3K27me3. However, to show that the method can identify cell type-specific signals, it is necessary to compare the samples D1 and D2 samples more directly. Using the presence or absence of a peak in one cell type versus another can lead to false positives. For example, a peak could be just below the threshold in one cell type and just above threshold in the other. Methods that directly compare read counts or signal across peaks should be applied to identify regions with a bias for D1 and D2 MSNs. Those differential regions should be associated with differential gene expression Ribo-Tag.

We thank the reviewer for their perspective on our initial analysis and agree that D1 and A2a samples must be compared directly to identify cell type specific signals. We previously normalized H3K4me3 and H3K27me3 read counts per million, assuming a consistent signal to noise ratio across replicates and samples (Supplementary Figure 2.2M, N). In the revised manuscript we assumed H3K4me3 and H3K27me3 read counts per million in A2a and D1 to be equal at genes whose expression level is also equal (See: Results, Page 9, Lines 214-216)¹. This new analysis allowed us to conclude, H3K27me3 and H3K4me3 enrichment is associated with cell type specific genes identified by Ribo-tag (Figure 3K-M).

The “poised” chromatin state has traditionally referred to more developmentally poised as opposed to poised for neural activity-dependent stimulation. For example, Kim et al 2010 (<https://pubmed.ncbi.nlm.nih.gov/20393465/>) noticed a depletion of H3K27me3 at enhancers that bind CBP in response to neural activity. Is there additional evidence to suggest that bivalent/co-enriched regions should be stimulus-dependent? Within this dataset are known IEGs or cell type specific IEGs enriched for co-occurring H3K27me3/H3K4me3?

Indeed, gene posing has traditionally been defined as developmentally poised, such that loss of bivalency permits transcription factor binding for cell type specific gene expression^{2,3}. We are intrigued by the concept that a similar mechanism may be at play in postmitotic neurons, as a mechanism for stable, long-term changes in gene expression downstream of activation signal (See: Introduction, Page 3-4, Lines 77-86). The revised manuscript includes additional evidence that loss of bivalent/co-enriched regions permits stimulus-dependent expression (See: Results, Page 10, Lines 246-256). These findings suggested H3K27me3 is an important factor for not only cell type specific expression but may also predict future gene activity in response to stimuli (Figure 4F, G; See: Discussion, Page 13, Lines 328-335).

Minor Issues

- Remove “149 wds” from abstract header

Removed (See: Introduction; Page 2; Lines 41).

- The “In contrast” header doesn’t seem to make sense, since both pieces of evidence support the role of MSNs in disease processes. (line 53)

Removed (See: Introduction; Page 2; Lines 48).

- Reference 23 (Skene and Henikoff) does not seem to address the increase sequencing noise

with FACS. In addition, identifying a cell type-specific nuclear protein and the right antibody are other major limitations of FACS.

We have revised the manuscript to clarify that Skene and Heinikoff 2017 does not support increased sequencing noise with FACS but rather the decreased noise with CUT&RUN compared to ChIP-Seq. We included references ⁴⁻⁸ to support our statement (See: Introduction; Page 3; Lines 69-73) that FACS can cause cellular stress and results in increased sequencing noise observed with INTACT -> FACS -> native ChIP-Seq ⁹. In addition, we included new analysis to support the claim that INTACT -> FACS -> native ChIP-Seq resulted in increased sequencing noise (See: Results; Page 7; Lines 175-181; Supplementary Figure 3C-F). With this additional analysis, we conclude ICuRuS provides consistent and reliable high quality epigenomic profiling data across different hPTMs.

- The methods mention “specificity” and yield, but Fig. 1 uses the terminology “purity’ and yield.

Figure 1E terminology was changed from purity to specificity (See: Methods; Page 16; Lines 83, 399-406).

- Fig1F and Fig1G – The statistical test applied and the p-values for the comparisons are not reported.

Statistics are now included (See: Figure Legends; Page 21; 529-533).

- Fig2D – The complete absence of any signal (or even noise) from the input is hard to believe. If true, why are the input values so different for the two antibodies

Antibody epitope recognition can explain differences in input values (See: Results, Page 7, Lines 169-172) ^{10,11}.

- Fig3 – It would be helpful if the direction arrows on the transcripts were larger so that the 5’ and 3’ ends of the genes can be easily identified.

We have included text for each track to ensure 5’ and 3’ ends can easily be identified.

Reviewer #2 (Remarks to the Author):

The study by Carpenter et al. combined cell-type specific nuclei isolation by the INTACT method with CUT&RUN in order to profile active (H3K4me3) and repressive (H3K27me3) histone modifications in two medium spiny neuron (MSN) subtypes of the mouse striatum, namely Adenosine 2a Receptor (A2a)- expressing and Dopamine Receptor D1 (D1)-expressing neurons. The authors first test the efficiency of the H3K4me3 and H3K27me3 CUT&RUN using the N2a cell line by comparing the CUT&RUN enrichment data to the available corresponding ChIP-seq data from the same cell line. Once the CUT&RUN protocol was established, they work with the INTACT-isolated A2a and D1 nuclei from the mouse brain and show comparable H3K4me3 and H3K27me3 enrichment data to the available corresponding ChIP-seq data from the bulk mouse nucleus accumbens brain tissue. The authors conclude that their INTACT-CUT&RUN protocol allows profiling of both active and repressive histone marks in a cell-type specific way in a brain tissue obtained from a single mouse, using only around 10,000 neuronal nuclei, which would not be permissive with current ChIP-seq protocols.

We thank the reviewer for acknowledging the importance of ICuRuS being performed on a single cell type from a single mouse. The development of this protocol enables future studies that investigate how individual differences in hPTM enrichment associate with variation in behavior.

They also highlight the advantages of not using FACS sorting (for cell type purification) or formaldehyde fixation (for histone modification profiling) in their current protocol as these procedures may lead to artifacts and increased background.

I believe that this is a very valuable tool for neuroscientists and would like to applaud the authors for their comprehensive protocol optimization and comparisons, including not only using both cell line and brain tissue but also multiple antibodies to test the efficiency of their protocol.

We thank the reviewer for recognizing our efforts to generate a robust protocol for profiling hPTMs in specific neuron types which has been notoriously difficult for the neuroscience community. Importantly, we appreciate the acknowledgement by this reviewer that the establishment of the protocol required extensive optimization, requiring time and effort comparable to our research on novel biology.

However, while this protocol is valuable, I believe that, by, this study does not offer any novelty in terms of biological insight into gene regulation in the brain and this is its major limitation. I would suggest that the authors expand on this study and use the developed protocol to answer a biological question regarding cell type-specific chromatin and gene regulation in striatal MSNs. In addition, I would like to offer a couple of comments that may be useful to the authors.

To improve the biological insights gleaned from this initial work, the revised manuscript includes new analyses that necessitate cell type specific hTPM and transcriptome profiling data. First, the revised manuscript directly compares hPTM enrichment across cell types (See: Results; Page 10; Lines 218-225), to identify cell-type specific modes of epigenetic regulation (See: Results; Page 10-11; Lines 258-271). Second, the revised manuscript analyzes the association between H3K4me3/H3K27me3 enrichment and cocaine-activated cell-type specific expression (See: Results; Page 10; Lines 246-258; Figure 4F, G). In conclusion, the revised manuscript is improved with respect to both its main utility as a novel method, and provides novel biological insights related to the cell-type specific epigenome.

We agree that the biological insights of this study are limited and have revised the abstract and introduction to highlight this manuscript more clearly as a resource for methodological advancement. The utility of this protocol to the research community warrants publication in Nature Communications because of its broad readership and history of publishing methodology focused articles that are similar in scope¹²⁻¹⁷.

To further its utility as a methods manuscript, the revised manuscript includes a comprehensive protocol (See: Protocol; Page 32-35; Lines 928-1101) and a rigorous analysis pipeline (See: Methods; Page 20; Lines 508-512) that advances our interpretations of cell type specific gene expression in brain.

1A I agree that FACS and formaldehyde crosslinking have their own limitations. However, the INTACT method requires using genetically modified animals. While these mice may not show gross changes in phenotype, subtle changes in mouse physiology may compromise some studies – I believe that the authors should discuss this.

We thank the reviewer for making this important point and we now include additional discussion to address limitations due to the use of transgenic animals (See: Discussion, Page 12, Lines 286-292). INTACT/ICuRuS require expression of SUN1GFP, but do not require constitutive expression of the SUN1GFP which may be used to avoid subtle changes in physiology. Although our protocol uses double transgenic mice, ICuRuS can be adapted for use with viral vectors expressing SUN1GFP in single transgenic Cre⁺ animals (including rats), and in non-transgenic animals using cell-type specific promoters. We have now included a discussion regarding the utility of ICuRuS in non-transgenic animals (See: Discussion, Page 12, Lines 288-293).

1B In addition, CUT&RUN may be efficient for histone modifications but less so for transcription factor binding (or this may require cross-linking too?). In general, native ChIP also does not require cross-linking and is a very efficient way of profiling histone modifications.

The benefit of CUT&RUN is that it is very efficient at profiling hPTMs and TFs at a low cost but does not require crosslinking¹⁸⁻²⁰. Although native ChIP is an efficient method for profiling hPTMs; it is not as efficient at profiling less abundant TF interactions given incomplete extraction efficiency of protein-DNA complexes and potential loss of binding. In fact, we found an improvement in H3K4me3 signal when comparing FACS -> native ChIP-seq from Mo et al 2015 to ICuRuS (Supplementary Figure 3C, D). Specifically, we found improved H3K4me3 signal to noise quantified by FRiP (Supplementary Figure 3C, E). H3K27me3 FRiP was more similar between the methods (Supplementary Figure 3D, F). However, background was higher in H3K27me3 ChIP-seq which increases the probability of false positives (Supplementary Figure 3F). These data provide evidence that ICuRuS was equal to or better than native ChIP-Seq although different nuclei isolation techniques were utilized. Namely, the required sequencing depth for CUT&RUN is 1/10 of the required reads for native ChIP-Seq which significantly reduces cost (See: Discussion, Page 11, Lines 279-285).

1C More generally, the authors should discuss, in more detail, both the advantages and disadvantages of their approach compared to ChIP-seq, which is still used as the standard histone modification profiling technique by many labs, in a more naturalistic setting.

We have discussed the advantages and disadvantages of CUT&RUN and ChIP-seq (See: Discussion, Page 12, Lines 294-304).

2) Since this is a very nice protocol that could be helpful to many research groups and the method development is the main contribution of this manuscript, I suggest that the authors provide a step-by-step protocol as part of their Supplementary Information. In addition, it would be nice to provide experimental quality control (QC) checkpoints that are used in the protocol to confirm that the procedure proceeds well such as bioanalyzer library prep traces, Qubit measurements, etc. Having a step-by-step protocol would likely allow for broader use of this method by other research groups.

A detailed protocol was included to allow for broader use of this method by other research groups (See: Protocol; Page 32-35; Lines 928-1101). We also included bioanalyzer traces (Supplementary Figure 2.2 K, L) which act as a checkpoint to confirm the protocol is working.

Reviewer #3 (Remarks to the Author):

This manuscript by Carpenter, Fischer, Zhang, et al describes a new protocol using INTACT and CUT&RUN (CnR) methods, together ICuRUS, to profile cell-type specific histone modifications in individual mouse brain tissue samples. Using these combined methods overcomes the limitations

of other neuro-epigenomic techniques such as ChIP-Seq, which cannot perform cell-type specific analysis or incorporate information on the individual animal's behavior or disease pathology due to the required high amount of starting material (i.e. pooling of samples). This method was then applied to profile 2 histone methylation marks, a marker of transcriptional activation (H3K4me3) and a marker of transcriptional repression (H3K27me3) within 2 striatal cell types, D1R-expressing and A2a-expressing cells, to determine how these marks regulate cell-type specific gene expression.

We thank the reviewer for highlighting the utility of ICuRuS for the broader neuroscience community. Significant data suggested individual variability is an important factor in brain diseases²⁵⁻²⁷. The major advance of our approach is the ability to perform hPTM profiling on a tissue from a single mouse, whereas most studies pool subjects and obscure individual differences in hPTM enrichment. Therefore, epigenetic profiling in individual mice will make significant advances in our ability to predict disease susceptibility.

The authors first demonstrated this protocol's ability to successfully isolate cell-type specific nuclei from the striatum of either D1R-Cre; or A2a-Cre; SUN1-GFP transgenic mice using RT-qPCR and immunohistochemistry. Separately, their CnR method using N2a cells produced high quality reads and similar distributions of both H3K4me3 and H3K27me3 reads in comparison to previously published N2a ChIP-Seq datasets. Following the validation of both separate approaches, the authors used ICuRuS to profile H3K4me3 and H3K27me3 in D1R- and A2a- nuclei from mouse striatal tissue. ICuRuS replicate datasets were highly reproducible, and overall displayed consistent pattern of read coverage to ChIP-Seq.

We thank the reviewer for noting the utility of our novel approach to improve over existing limitations, and the extensive validation applied in development of the protocol.

To compare the relationship between cell-type specific histone marks and mRNA expression, the authors next overlaid their ICuRuS datasets with previously published D1R and D2R-specific transcriptome datasets. As expected, cell-type specific H3K4me3 reads were enriched in genes highly expressed in both D1R and A2a as H3K27me3 were enriched in lowly expressed genes. Finally, the combinatorial function of H3K4me3 and H3K27me3 were investigated to determine whether a subset of genes enriched for both marks were poised for high expression upon stimulation. Potential higher sensitivity for detecting co-enrichment was seen from CnR and ICuRuS datasets versus ChIP-Seq. Majority of co-enriched domains from ICuRuS data were seen in both A2a and D1 MSNs, and these genes were associated with processes relating to cell differentiation, development.

We thank the reviewer for noting that our novel approach allows improved analysis of co-enrichment of hPTMs.

We agree immediate early genes (IEGs) did not display a consistent pattern of co-enrichment and genes expressed within a specific cell-types did not display a co-enrichment pattern specific to a cell-types. We interpret this as a novel finding with implications on cell type specific gene transcription.

We appreciate this comment and agree that we defined an unexpected pattern of cell-type specific mRNA and hPTMs. The revised manuscript addresses this with additional data (See: Results; Page 10-11; Lines 246-271), analyses (See: Results; Page 9; Lines 214-218) and clarifications (See: Discussion; Page 13; Lines 326-337). Briefly, we (1) found that ICuRuS improved sensitivity in detecting co-enrichment (Supplementary Figure 3C-F) (2) directly compared hPTM profile and

gene expression between the two MSN subtypes (Figure 3L-N). This new analysis allowed us to conclude that (1) immediate early genes (IEGs) expressed in A2a and D1 nuclei displayed a consistent pattern of H3K27me3 and H3K4me3 enrichment and (2) genes upregulated in a A2a and D1 by cocaine display H3K27me3 and H3K4me3 enrichment pattern specific to cell-types.

Overall, the authors present a thorough validation of their protocol which optimizes the combined use of 2 previously established methods for cell-type specific epigenetic profiling. Their use of previously published datasets to analyze and confirm the accuracy of this protocol is also well-performed.

We thank the reviewer for noting our rigorous and extensive optimization and validations by comparison to published datasets.

However, this analysis produces limited new conclusions for the field.

To improve the biological insights gleaned from this initial work, the revised manuscript includes new analyses that necessitate cell type specific hTPM and transcriptome profiling data in individual mice. First, the revised manuscript directly compares hPTM enrichment across cell types, to identify cell-type specific modes of epigenetic regulation (See: Results; Page 10-11; Lines 246-271). Second, the revised manuscript analyzes how H3K4me3, H3K27me3, and their co-enrichment regulate cocaine-activated cell-type specific expression (See: Results; Page 10-11; Lines 260-271; Supplementary Figure 4A). In conclusion, the revised manuscript is improved with respect to both its main utility as a novel method, and provides novel biological insights related to the cell-type specific epigenome.

We agree that the biological insights of this study are limited and have revised the abstract (See: Abstract; Page 2; Lines 33-35) and introduction (See: Introduction; Page 2-3; Lines 42-55) to highlight this manuscript more clearly as a resource for methodological advancement. The utility of this protocol to the research community warrants publication in Nature Communications because of its broad readership and history of publishing ¹²⁻¹⁷ methodology focused articles that are similar in scope.

To further its utility as a methods manuscript, the revised manuscript includes a comprehensive protocol (See: Protocol; Page 32-35; Lines 928-1101) and a rigorous analysis pipeline (See: Methods; Page 20; Lines 510-514) that advances our interpretations of cell type specific gene expression in brain.

For instance, their results confirm that well established repressive and permissive histone marks correlate with low and high gene expression respectively in these striatal cell types at baseline.

We thank the reviewer for noting our rigorous validation of canonical roles of hPTMs and agree that while these analyses are necessary for development of a cell-type specific protocol, they did not provide novel biological insights.

When examining the potential role of co-enrichment of these histone marks on gene expression to poise gene expression, majority of the co-enriched domains were common in both cell types, indicating a lack of genes that are poised for expression in a cell-type specific manner via these histone marks.

We agree there is a lack of genes that are poised for expression in a cell-type specific manner via these histone marks, but we interpret this data as a novel finding. Although most of the co-

enriched domains were common in both cell types, a small portion of genes were co-enriched exclusively in A2a or D1 (Figure 4D; D1-specific 4%, A2a-specific 44%), which was consistent with the identification of very few cell type specific genes²⁸ and activity dependent cell type specific genes²⁹. As expected, these genes included cell type specific genes (See: Results; Page 8; Lines 185-193; Figure A-F) and genes activated in a cell type specific manner in response to cocaine (See: Results; Page 10-11; Lines 260-271; Supplementary Figure 4A).

Further, when investigating co-enrichment of histone marks on immediate early genes, which the authors suggested would likely be poised in either cell type for responses to stimulation, no pattern was observed.

We agree that we initially presented sparse data on cell type specific activation and no clear pattern was observed. The revised manuscript includes additional data that identified a pattern in H3K4me3 and H3K27me3 enrichment at the top 10 differentially expressed genes (90% of genes were IEGs) in A2a or D1 nuclei following cocaine exposure (See: Results; Page 10; Lines 242-258; Figure 4F, G). Specifically, we found expression of cell type specific, and cocaine activated genes are depleted in H3K27me3, which provides new conclusions regarding specific hPTMs that sustain transcriptional regulation in specific cell types (See: Discussion, Page 13, Lines 328-335). In this way, neuronal subtype specific expression was conferred by either loss of a repressive hPTM in the expressing subtype.

This manuscript presents a useful new protocol for cell-type specific epigenetic profiling; however, we provide suggestions below to enhance its significance to field.

1) It is appreciated that the authors show mRNA enrichment or depletion of A2A and D1 in Figure 1 and H3K4me3 or H3K27me3 enrichment on A2A or D1 promoters in Figure 3. However, the study would benefit from further cell type specific validation including enrichment of H3K4me3 or H3K27me3 on promoters of other known enriched genes (i.e. A2A MSNs- D2, Penk, Gpr6; D1 MSNs- Pdyn, Tac1, Chrm4). This would be especially important since the enrichment in A2A and D2 promoters may include the BAC transgene which uses these promoters.

We thank the reviewer for suggesting these target genes. We agree that further validation of A2a- and D1-specific genes is important given that the BAC transgene uses A2a and Drd1 promoters. The revised manuscript now additionally includes analysis of hPTMs in each cell type at *Penk*, *Tac1* and *Pdyn*, alongside *A2a*, *Drd2*, and *Drd1* (See: Results; Page 8; Lines 185-193; Figure 3A-F).

2) To use this method to further investigate the role of combinatorial histone marks on poising gene expression, the authors could perform additional experiments to identify whether these combined marks or others are critical regulators. Considering the expertise of the authors, this could be performed by examining whether chronic cocaine exposure impacts the co-enrichment of these histone marks on immediate early genes within D1 or A2A striatal cell types. Cocaine exposure can induce differential effects on IEG expression in these cell types, therefore this treatment may better parse apart whether these combined marks poise future gene expression based on prior experience. In addition, the effects of cocaine on cell-type specific histone modifications would yield highly useful data for the field as this could be overlaid with cell-type specific RNA-Seq data sets.

We appreciate the reviewer's enthusiasm for the utility of our approach in further investigation of poised gene expression. We agree our data has the potential to identify the role of cell type specific H3K4me3/H3K27me3 co-enrichment on gene activation. The revised manuscript

addresses this question at a set of MSN-subtype specific cocaine-activated genes (See: Results; Page 10; Lines 246-258). Intriguingly, we found genes activated by cocaine have high H3K4me3 enrichment and low H3K27me3 enrichment, which suggested that H3K27me3 negatively regulates cocaine gene activation (Figure 4F, G). We applied a similar analysis to *Egr3*, a known regulator of cocaine reward behavior (See: Results; Page 10-11; Lines 256-272; Supplementary Figure 4A). Overall, our current data is highly useful for generating novel hypotheses regarding the factors that mediate cocaine induced expression.

3) The authors also compare their protocol to other isolation methods such as FACS and emphasize how their technique prevents false signals. Direct comparisons of those approaches would further demonstrate the importance of their protocol and be critical information to the field.

We agree with the reviewer that it is important to directly compare FACS and INTACT³⁰ (See: Introduction; Page 3; Lines 56-76). The revised manuscript includes comparisons between ICuRuS and INTACT ChIP-Seq (See: Results; Page 7; Lines 175-181; Supplementary Figure 3C-F). Briefly, we found that ICuRuS improved H3K4me3 signal to noise compared to ChIP-seq quantified by FRiP, whereas H3K27me3 signal was similar across the methods. However, background was higher in H3K27me3 ChIP-seq which increases the probability of false positives. With this additional analysis, we concluded ICuRuS provides consistent and reliable high quality epigenomic profiling data across different hPTMs.

4) Many studies identify distinct behavioral outcomes through manipulating MSN subtypes across striatal regions (i.e. dorsal lateral striatum, dorsal medial striatum, and different regions of the nucleus accumbens). Presumably these regions have distinct epigenetic/transcriptome adaptations that correlate with behavioral outcomes. The current study isolates total striatum for cell type specific analysis. Do the authors think they can perform the histone post translational modification profiles in these striatal regions or are they still limited by material especially in smaller nucleus accumbens regions? The authors might include some discussion on this in the discussion See.

We thank the reviewer for asking this important question which has implications on the utility of ICuRuS. Yes, CnR can be performed on small amounts of material including single cells which facilitates hPTM profiling in rare cell types and smaller sub brain regions of brain^{18,31}. The revised manuscript includes discussion of this topic (See: Discussion, Page 12, Lines 295-296).

R2R References

1. Polit, L. *et al.* CHIPIN: ChIP-seq inter-sample normalization based on signal invariance across transcriptionally constant genes. *BMC Bioinformatics* **22**, 407 (2021).
2. Gao, Y., Gan, H., Lou, Z. & Zhang, Z. Asf1a resolves bivalent chromatin domains for the induction of lineage-specific genes during mouse embryonic stem cell differentiation. *Proc. Natl. Acad. Sci. U. S. A.* **115**, E6162–E6171 (2018).
3. Kim, T.-K. *et al.* Widespread transcription at neuronal activity-regulated enhancers. *Nature* **465**, 182–187 (2010).
4. Andrä, I. *et al.* An Evaluation of T-Cell Functionality After Flow Cytometry Sorting Revealed p38 MAPK Activation. *Cytometry A* **97**, 171–183 (2020).
5. Binek, A. *et al.* Flow Cytometry Has a Significant Impact on the Cellular Metabolome. *J. Proteome Res.* acs.jproteome.8b00472 (2018) doi:10.1021/acs.jproteome.8b00472.
6. Beliakova-Bethell, N. *et al.* The effect of cell subset isolation method on gene expression in leukocytes. *Cytom. Part J. Int. Soc. Anal. Cytol.* **85**, 94–104 (2014).
7. Mollet, M., Godoy-Silva, R., Berdugo, C. & Chalmers, J. J. Acute hydrodynamic forces and apoptosis: A complex question. *Biotechnol. Bioeng.* **98**, 772–788 (2007).
8. Mollet, M., Godoy-Silva, R., Berdugo, C. & Chalmers, J. J. Computer simulations of the energy dissipation rate in a fluorescence-activated cell sorter: Implications to cells. *Biotechnol. Bioeng.* **100**, 260–272 (2008).
9. Mo, A. *et al.* Epigenomic Signatures of Neuronal Diversity in the Mammalian Brain. *Neuron* **86**, 1369–1384 (2015).
10. Busby, M. *et al.* Systematic comparison of monoclonal versus polyclonal antibodies for mapping histone modifications by ChIP-seq. *Epigenetics Chromatin* **9**, 49 (2016).
11. Rothbart, S. B. *et al.* An Interactive Database for the Assessment of Histone Antibody Specificity. *Mol. Cell* **59**, 502–511 (2015).
12. Kaya-Okur, H. S. *et al.* CUT&Tag for efficient epigenomic profiling of small samples and single cells. *Nat. Commun.* **10**, 1930 (2019).
13. Lal, A. *et al.* Deep learning-based enhancement of epigenomics data with AtacWorks. *Nat. Commun.* **12**, 1507 (2021).
14. Brind'Amour, J. *et al.* An ultra-low-input native ChIP-seq protocol for genome-wide profiling of rare cell populations. *Nat. Commun.* **6**, 6033 (2015).
15. Rossi, M. J., Lai, W. K. M. & Pugh, B. F. Simplified ChIP-exo assays. *Nat. Commun.* **9**, 2842 (2018).
16. Carter, B. *et al.* Mapping histone modifications in low cell number and single cells using antibody-guided chromatin tagmentation (ACT-seq). *Nat. Commun.* **10**, 3747 (2019).
17. Hu, B. *et al.* Neuronal and glial 3D chromatin architecture informs the cellular etiology of brain disorders. *Nat. Commun.* **12**, 3968 (2021).
18. Patty, B. J. & Hainer, S. J. Transcription factor chromatin profiling genome-wide using uliCUT&RUN in single cells and individual blastocysts. *Nat. Protoc.* **16**, 2633–2666 (2021).
19. Hainer, S. J., Bošković, A., McCannell, K. N., Rando, O. J. & Fazio, T. G. Profiling of Pluripotency Factors in Single Cells and Early Embryos. *Cell* **177**, 1319–1329.e11 (2019).
20. Skene, P. J., Henikoff, J. G. & Henikoff, S. Targeted in situ genome-wide profiling with high efficiency for low cell numbers. *Nat. Protoc.* **13**, 1006–1019 (2018).
21. Liu, N. *et al.* Direct Promoter Repression by BCL11A Controls the Fetal to Adult Hemoglobin Switch. *Cell* **173**, 430–442.e17 (2018).
22. Liu, N. *et al.* Transcription factor competition at the γ -globin promoters controls hemoglobin switching. *Nat. Genet.* **53**, 511–520 (2021).
23. Stroud, H. *et al.* An Activity-Mediated Transition in Transcription in Early Postnatal Neurons. *Neuron* **107**, 874–890.e8 (2020).

24. Skene, P. J. & Henikoff, S. An efficient targeted nuclease strategy for high-resolution mapping of DNA binding sites. *eLife* **6**, e21856 (2017).
25. Sapolsky, R. M. Stress and the brain: individual variability and the inverted-U. *Nat. Neurosci.* **18**, 1344–1346 (2015).
26. MacDonald, S. W. S., Nyberg, L. & Bäckman, L. Intra-individual variability in behavior: links to brain structure, neurotransmission and neuronal activity. *Trends Neurosci.* **29**, 474–480 (2006).
27. Van Horn, J. D., Grafton, S. T. & Miller, M. B. Individual Variability in Brain Activity: A Nuisance or an Opportunity? *Brain Imaging Behav.* **2**, 327–334 (2008).
28. Kronman, H. *et al.* Biology and Bias in Cell Type-Specific RNAseq of Nucleus Accumbens Medium Spiny Neurons. *Sci. Rep.* **9**, 8350 (2019).
29. Savell, K. E. *et al.* A dopamine-induced gene expression signature regulates neuronal function and cocaine response. *Sci. Adv.* **6**, eaba4221 (2020).
30. Chongtham, M. C., Butto, T., Mungikar, K., Gerber, S. & Winter, J. INTACT vs. FANS for Cell-Type-Specific Nuclei Sorting: A Comprehensive Qualitative and Quantitative Comparison. *Int. J. Mol. Sci.* **22**, 5335 (2021).
31. Hainer, S. J. & Fazio, T. G. High-Resolution Chromatin Profiling Using CUT&RUN. *Curr. Protoc. Mol. Biol.* **126**, e85 (2019).

Reviewers' Comments:

Reviewer #1:

Remarks to the Author:

The major issue of a lack of direct D1 and A2A comparison is addressed with figure 3K-M.

The major issue of the relationship between H3K27me3 and IEG expression has also been resolved by a much more nuanced analysis of cocaine-dependent gene expression and the profiled histone modifications. The new results of the new analysis provide compelling evidence that the presence of H3K27me3 can impede the ability of neural activity to drive cell type-specific gene expression.

The minor comments were also addressed.

Overall, the analysis of the manuscript is more straightforward and there is more biological novelty presented. The methodological strengths noted by myself and other reviewers are not diminished. I strongly recommend for publication.

Reviewer #2:

Remarks to the Author:

I appreciate all the revisions that the authors incorporated in the new version of the manuscript as well as that they provided the complete protocol. I believe this is a fine method that generates data of good quality with cell type-specific resolution and appreciate authors' efforts to prove this and provide a new method for the field.

However, I still believe that this manuscript lacks significant biological insights and this has not been well addressed in the revision. The authors attempted to address this point by integrating cell type specific gene expression data from control- and cocaine-treated rodents with their ICuRuS-generated data. The majority of the data they show are expected -- the enrichment of H3K4me3 is associated with active genes; the enrichment of H3K27me3 is associated with inactive genes. It is nice that the authors can show cell type specificity of hPTMs using their method, but there is no new biological insight here. The bivalency data are also not conclusive. Finally, the attempt to connect their data with cocaine-induced gene expression was not convincing. For instance, in Fig 4L, a hardly visible H3K27me3 enrichment in the vicinity of Egr3 in A2a cells (vs no enrichment of H3K27me3 in Egr3-inducing D1 cells) is too weak to support authors' statement that "H3K27me3 depletion regulates cell-type specific gene activity in response to stimuli in A2a and D1 MSNs". This is also not consistent with visible H3K27me3 enrichment in both cell types around A2a-specific Cartpt gene (Fig 4M).

Simply, the authors overstate the biological significance of their data and this is how this culminates in their discussion:

"Together, these analyses answered long standing questions in the field regarding the factors that regulate and sustain activity dependent transcription in specific cell populations within reward circuitry."

I believe that much more data are needed to support authors' conclusions and the paper is just not solid in its current form. The authors mention some of their unpublished data (and I suggest that these statements are removed if the data are not shown) and also rightly claim that they are in a great position to answer some of the important questions that they started to address in this manuscript. Thus, I suggest that the authors either provide more data that will show how their method brings new biological insights into the field or publish this nice method in a different form.

Reviewer #3:

Remarks to the Author:

The authors have addressed all concerns. They have provided new data that enhances the significance of this study. This study will be an important resource for cell type specific epigenetic

profiling in the brain.

REVIEWER COMMENTS

Reviewer #1 (Remarks to the Author):

The major issue of a lack of direct D1 and A2A comparison is addressed with figure 3K-M.

We agree that the new data provided in Figure 3K-M provides strong evidence that ICuRuS produces good quality data.

The major issue of the relationship between H3K27me3 and IEG expression has also been resolved by a much more nuanced analysis of cocaine-dependent gene expression and the profiled histone modifications. The new results of the new analysis provide compelling evidence that the presence of H3K27me3 can impede the ability of neural activity to drive cell type-specific gene expression.

We thank the reviewer for the suggestion to further define the relationship between H3K27me3 and IEG expression. We agree that our data supports the conclusion H3K27me3 enrichment is one possible mechanism that impedes cell type-specific expression of some genes.

The minor comments were also addressed.

We thank the reviewer for recognizing our effort to address all concerns.

Overall, the analysis of the manuscript is more straightforward and there is more biological novelty presented. The methodological strengths noted by myself and other reviewers are not diminished. I strongly recommend for publication.

We thank the reviewer for noting we have provided sufficient biological novelty for publication. We agree that our data can be used to generate compelling hypothesis regarding epigenetic mechanisms at cell type specific genes.

Reviewer #2 (Remarks to the Author):

I appreciate all the revisions that the authors incorporated in the new version of the manuscript as well as that they provided the complete protocol.

We thank the reviewer for acknowledging the extent of our revisions and for providing the complete protocol. We agree this protocol is valuable to the neuroscience field.

I believe this is a fine method that generates data of good quality with cell type-specific resolution and appreciate authors' efforts to prove this and provide a new method for the field.

We thank the reviewer for pointing out this important distinction. We have included new data analyzing H3K4me3 and H3K27me3 peaks in A2a and D1 nuclei. We found H3K4me3 peaks are

associated with most cell type specific genes expressed in the respective cell type, whereas few genes were enriched in H3K27me3 in both A2a and D1 nuclei (See: Results, Page 8, Line 211-215; Figure 3G and H). These data suggested that H3K4me3 is a poor predictor of cell type specific gene expression, whereas, H3K27me3 depletion may be associated with cell type specific gene expression. We now highlight H3K27me3 depletion at MSN-specific genes as a possible mechanism for MSN specific gene expression (See: Discussion, Page 14, Lines 392-402). These data provide valuable evidence regarding specific genomic regions amenable to cell type specific H3K27me3 editing.

However, I still believe that this manuscript lacks significant biological insights and this has not been well addressed in the revision.

To uncover mechanisms of cell-type specific gene activation, we defined hPTM enrichment at cocaine-activated genes specific to each MSN subtype. We discovered *Penk* was depleted in H3K27me3 at baseline and upregulated following cocaine exposure in A2a neurons relative to D1 neurons. Similarly, *Tac1* was depleted in H3K27me3 at baseline and upregulated following cocaine exposure in D1 neurons relative to A2a neurons. However, the role of *Penk* and *Tac1* in cocaine reward has not been investigated in specific cell types. Thus, we focused on *Egr3* which showed cell type specific effects on cocaine reward. *Egr3* showed D1-specific depletion of H3K27me3, while H3K4me3 remained enriched in both cell types, and cocaine dependent activation. Taken together, hPTM profiling by ICuRuS is of sufficient resolution for the analysis of cell-type hPTM enrichment in brain (See: Results, Page 11, Lines 303-307).

The authors attempted to address this point by integrating cell type specific gene expression data from control- and cocaine-treated rodents with their ICuRuS-generated data. The majority of the data they show are expected -- the enrichment of H3K4me3 is associated with active genes; the enrichment of H3K27me3 is associated with inactive genes. It is nice that the authors can show cell type specificity of hPTMs using their method...

We appreciate acknowledgement of our rigorous validation. The unique data in this case is that the canonical role of hPTMs holds in specific neuronal subtypes - this analysis has not been performed previously. While existing knowledge about the role of hPTMs in bulk neuronal tissue informed our hypotheses, we are excited to have rigorously supported this hypothesis in our study of D1 and A2a neurons.

, but there is no new biological insight here.

The main biological insight is found when we consider cocaine-activation of cell-type specific genes. There is not yet a consensus as to the epigenetic mechanism of cocaine-activated gene expression in bulk or cell-type specific tissue. Given the critical role of H3K4me3 in specifying basal gene expression, we hypothesized enrichment of this modification would correlate with cocaine-activation. Yet we found that depletion of the repressive modification, H3K27me3, was associated with cell-type specific cocaine gene activation in many instances, such as at, *Penk*, *Tac1* and *Egr3*. To address the reviewer concerns, we have modified the text to tone down

instances where we extrapolated this to a more general mechanism (See: Discussion, Page 14, Lines 392-402).

The bivalency data are also not conclusive.

The major advance of this publication is the method ICuRuS, which provides good quality hPTM profiling data with cell type-specific resolution. Our lab investigated cocaine and bivalency more extensively in a recent publication ¹.

Finally, the attempt to connect their data with cocaine-induced gene expression was not convincing. For instance, in Fig 4L, a hardly visible H3K27me3 enrichment in the vicinity of *Egr3* in A2a cells (vs no enrichment of H3K27me3 in *Egr3*-inducing D1 cells) is too weak to support authors' statement that "H3K27me3 depletion regulates cell-type specific gene activity in response to stimuli in A2a and D1 MSNs".

We agree, the abundance of the H3K27me3 at *Egr3* is difficult to observe visually and it is not feasible to visually quantify hPTMs in the genome. Thus, we quantified the number of reads normalized to counts per million in a window 2 kb upstream and 1 kb downstream of the *Egr3* TSS (**Supplementary Figure 5A**). In addition, we observe H3K27me3 depletion at baseline and cocaine induced activation of A2a-specific *Penk* and D1-specific *Tac1* (See *Penk* and *Tac1* Figure 3), which provides evidence this mechanism extends to some genes, but we did not extend this investigation to all genes (See: Discussion, Page 14, Lines 392-402).

This is also not consistent with visible H3K27me3 enrichment in both cell types around A2a-specific *Cartpt* gene (Fig 4M).

We have clarified language (See: Discussion, Page 14, Lines 392-402) to specify our finding at *Egr3* is not universal to all genes. We state (See: Discussion, Page 13, Lines 357-359) that in some cases, such as *Cartpt*, neither H3K4me3 nor H3K27me3 are sufficient to specify cell-type specific expression and other mods are likely involved. Finally, we directly state we do not uncover the mechanism of cocaine dependent gene expression (See: Results, Page 11, Lines 291)

Simply, the authors overstate the biological significance of their data and this is how this culminates in their discussion:

"Together, these analyses answered long standing questions in the field regarding the factors that regulate and sustain activity dependent transcription in specific cell populations within reward circuitry."

We have modified the text to tone down instances where we extrapolated this to additional genes (See: Discussion, Page 14, Lines 392-402) or overstate the biological significance of our findings (See: Discussion, Page 14, Lines 401-402; See: Results, Page 11, Lines 291)

I believe that much more data are needed to support authors' conclusions and the paper is just not solid in its current form.

We agree that more data is required to support some of our previous conclusions. We now focus our conclusions on the quality of our ICuRuS data which facilitated the study of cell type specific epigenomic gene regulation in brain (See: Discussion, Page 14, Lines 403-406).

The authors mention some of their unpublished data (and I suggest that these statements are removed if the data are not shown) and also rightly claim that they are in a great position to answer some of the important questions that they started to address in this manuscript. Thus, I suggest that the authors either provide more data that will show how their method brings new biological insights into the field or publish this nice method in a different form.

We agree we are in a great position to answer some of the important questions related to cell type specific gene regulation. To publish this article in a different form, we have removed all references to unpublished data (See: Discussion, Page 12, Lines 326; Page 13, Lines 352) and focused our conclusions the quality of our ICuRuS cell type specific profiling (See: Discussion, Page 14, Lines 403-406).

Reviewer #3 (Remarks to the Author):

The authors have addressed all concerns. They have provided new data that enhances the significance of this study. This study will be an important resource for cell type specific epigenetic profiling in the brain.

We thank the reviewer for their helpful comments which have strengthened the significance of this study.

References

1. Fischer, D. K., Krick, K. S., Han, C., Woolf, M. T. & Heller, E. A. Cocaine regulation of Nr4a1 chromatin bivalency and mRNA in male and female mice. *Sci. Rep.* **12**, 15735 (2022).

Reviewers' Comments:

Reviewer #2:

Remarks to the Author:

The authors responded well to my comments. The manuscript parts that involved over-interpretation of the data are removed and the language of the manuscript is now well-balanced and corresponds with the data.

I just noticed a couple of inconsistencies in the manuscript that the authors should check:

1) On Page 7 of the main text, the authors are referring to Suppl Figure 2M-T. I don't believe that there is Suppl Figure 2T.

2) On page 10, Figures 4H-J are not mentioned in the main text.

3) Line 303-304 (and also in Discussion): the authors state that H3K27me3 signal in the vicinity of the Egr3 gene is depleted in A2a MSNs relative to D1 MSNs (Fig 4L). It appears to me that opposite is true: the H3K27me3 signal is depleted in D1 MSNs relative to A2a cells, which corresponds to cocaine-induced Egr3 activation specific to D1 neurons. I suggest clarifying this in the main text.

4) Line 361: replace 'mods' with 'modifications'

Reviewer #2 (Remarks to the Author):

The authors responded well to my comments. The manuscript parts that involved over-interpretation of the data are removed and the language of the manuscript is now well-balanced and corresponds with the data.

I just noticed a couple of inconsistencies in the manuscript that the authors should check:

1) On Page 7 of the main text, the authors are referring to Suppl Figure 2M-T. I don't believe that there is Suppl Figure 2T.

We have updated the reference to Supplemental Figure 2M-R

2) On page 10, Figures 4H-J are not mentioned in the main text.

We have revised the text to mention Figures 4H-J

3) Line 303-304 (and also in Discussion): the authors state that H3K27me3 signal in the vicinity of the Egr3 gene is depleted in A2a MSNs relative to D1 MSNs (Fig 4L). It appears to me that opposite is true: the H3K27me3 signal is depleted in D1 MSNs relative to A2a cells, which corresponds to cocaine-induced Egr3 activation specific to D1 neurons. I suggest clarifying this in the main text.

We thank the reviewer for finding these inconsistencies. Indeed, H3K27me3 signal in the vicinity of the Egr3 gene is depleted in D1 MSNs relative to A2a MSNs.

4) Line 361: replace 'mods' with 'modifications'

We replaced 'mods' with modifications